# Label-free single-cell RNA multiplexing leveraging genetic variability

Konrad Hoeft [1,2,7], Tore Bleckwehl [1,7], David Schumacher[1,3], Hyojin Kim[1], Robert Meyer [4], Qingqing Long[1], Ling Zhang[1], Christian Möller[1,5], Marian C. Clahsen-van Groningen [1,6], Anne Babler[1], Turgay Saritas [1,2], Ingo Kurth [4], Hendrik Milting[5], Sikander Hayat [1,8] ✉ & Rafael Kramann[1,2,8] ✉

Single cell RNA sequencing has provided unprecedented insights into the molecular cues and cellular heterogeneity underlying human disease. However, the high costs and complexity of single cell methods remain a major obstacle for generating large-scale human cohorts. Here, we compare current state-of-the-art single cell multiplexing technologies, and provide a widely applicable demultiplexing method, *SoupLadle*, that enables simple, yet robust high-throughput multiplexing leveraging genetic variability of patients.

The rapid rise of single-cell RNA sequencing (scRNA-seq) has provided unprecedented insights into the molecular and cellular cues shaping disease[1]. Unfortunately, the high cost of this technology remains a major limitation for the generation of data from larger human cohorts. However, the multifactorial origin and plasticity of many diseases, as well as high inter-individual differences in humans necessitate the generation of large-scale cohorts to pinpoint the elusive molecular drivers of disease[2]. Moreover, a highly labile transcriptome, which is susceptible to degradation or contamination during isolation, can lead to strong batch effects between experiments. To this end, multiplexing technologies offer an elegant solution, reducing both experimental batch effects and costs. Among the most common multiplexing-methods used are cell-labeling approaches (*CellPlex* or *Hashtagging*), where cells are tagged with a sample-specific oligonucleotide prior to pooling[3]. Alternatively, cells can be multiplexed by calling patient-specific SNPs from scRNA-seq data (*Vireo*, *Souporcell* or *Demuxlet*)[4–8]. At baseline however (without reference whole exome sequencing or bulkRNA-seq single nucleotide polymorphism (SNP) data), SNP-calling methods only discriminate (from here on referred to as deconvolute), but do not re-assign cells to patients due to the lack of information on reference patient-defining SNPs. Here, Vireo offers

demultiplexing of patients by integrating SNP data from bulkRNA or Whole Exome Sequencing (WES).

Here, we benchmark current state-of-the-art single-cell multiplexing technologies and, based on our results, present an improved genomic multiplexing framework termed *SoupLadle*.

## Results

### Benchmarking of scRNA-seq multiplexing methods

First, we benchmarked current cell labeling and genomic demultiplexing methods. To compare multiplexing methods, we isolated PBMC from five patients, labeled them with a patient-unique CellPlex-Oligo, and subsequently isolated a unique PBMC population for each patient using Fluorescence-Activated Cell Sorting (FACS) and gold-standard cell-population specific cell-surface markers (Fig. 1a, Supplementary Fig. 1a). We reasoned that this will lead to transcriptionally distinct cell populations for each patient, enabling us to benchmark multiplexing strategies against scRNA clustering as a reference. After FACS, samples were pooled for 10X 3′ scRNA-seq. For genomic demultiplexing, we performed bulkRNA and WES of each patient (Supplementary Fig. 1b, c).

After quality control, clustering clearly distinguished the five sorted PBMC populations, which we subsequently used as a reference

[1]Department of Medicine 2 (Nephrology, Rheumatology, Clinical Immunology and Hypertension), Medical Faculty, RWTH Aachen University, Aachen, Germany. [2]Department of Internal Medicine, Nephrology and Transplantation, Erasmus Medical Center, Rotterdam, The Netherlands. [3]Department of Anesthesiology, Medical Faculty, RWTH Aachen University, Aachen, Germany. [4]Institute for Human Genetics and Genomic Medicine, Medical Faculty, RWTH Aachen University, Aachen, Germany. [5]Erich and Hanna Klessmann Institute for Cardiovascular Research and Development, Clinic for Thoracic and Cardiovascular Surgery, Heart and Diabetes Center NRW, Bad Oeynhausen, Germany. [6]Department of Pathology and Clinical Bioinformatics, Erasmus MC, University Medical Center Rotterdam, Rotterdam, The Netherlands. [7]These authors contributed equally: Konrad Hoeft, Tore Bleckwehl. [8]These authors jointly supervised this work: Sikander Hayat, Rafael Kramann. ✉e-mail: shayat@ukaachen.de; rkramann@ukaachen.de

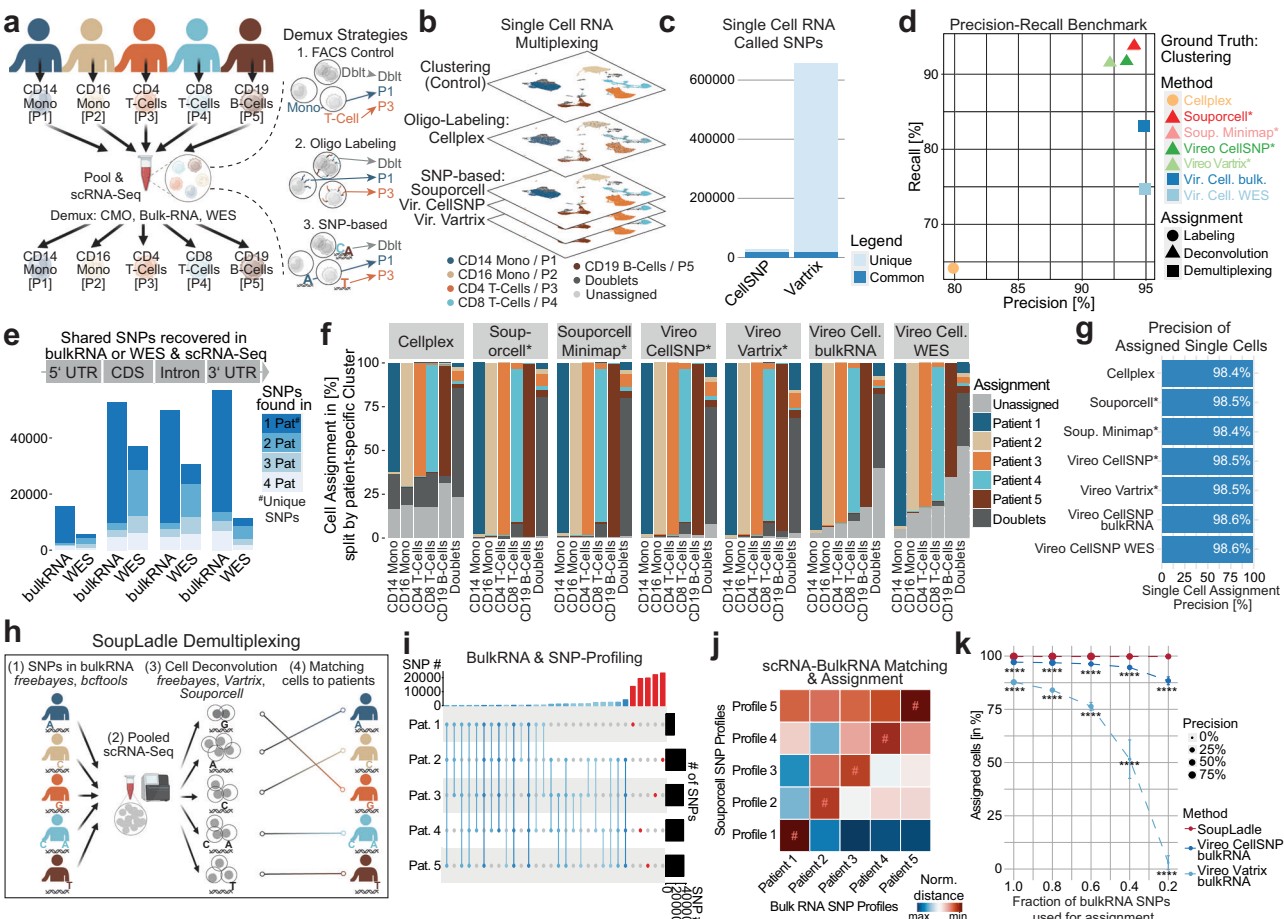

**Fig. 1 | Benchmarking of multiplexing methods in a PBMC single cell RNA sequencing dataset. a** Experimental design of PBMC multiplexing experiment. Mono Monocytes, Dblt Doublet, CMO Cell Multiplexing Oligo. The schematic drawing was created with BioRender (BioRender.com). **b** Conceptual cartoon showing UMAP representations of PBMC stratified by clustering or patient assignment by multiplexing methods. Vir Vireo. P Patient. **c** Quantification of SNPs called from scRNA-seq data using CellSNP or Vartrix. Common: Found with both methods. **d** Precision-Recall of multiplexing methods. Clustering is used as a reference. Soup Souporcell, Vir. Vireo, bulk bulkRNA, Cell. CellSNP. **e** Quantification of scRNA-seq SNPs recovered with BulkRNA or WES, split by genomic region. CDS coding sequence, UTR untranslated region. **f** Cell-to-patient assignment for each method stratified by clusters. **g** Precision of assigned single cells (excluding doublets) for each method. **h** Schematic design of SoupLadle framework. The

schematic drawing was created with BioRender (BioRender.com). **i** Assessment of patient-unique SNPs and intersections using PBMC bulkRNA-seq data. **j** Heatmap of normalized hamming distance for best-matching patient assignment with SoupLadle of scRNA-seq PBMC SNP profiles to patient bulkRNA-SNP profiles. SNP profiles were assigned to patients with the least normalized distance (labeled with #). **k** Assigned cells [in %] and precision with continuously downsampled ($n = 10$ per condition) bulkRNA-SNP numbers for SoupLadle, Vireo CellSNP bulkRNA, and Vireo Vartrix bulkRNA in PBMC dataset. Data are presented as mean assigned cells ±Standard Deviation. For (**k**) a Wilcoxon signed-rank Test was performed. Source data and exact *p* values are provided in the Source Data file. Exact numbers for cell sorting, and cell demultiplexing stratified by patient for each multiplexing method are provided in Supplementary Data 2. ****$p < 0.0001$. *Vireo CellSNP, Vireo Vartrix, and Souporcell were assigned to patients based on overlap with clustering.

to benchmark multiplexing methods (Fig. 1b, Supplementary Fig. 1d–f). SNPs were quantified as recommended using *CellSNP* for Vireo and either *Vartrix* or *Minimap* for Souporcell (Souporcell or Souporcell Minimap). As Vartrix recovered considerably more SNPs than CellSNP due to stricter filtering (i.e., minimum of 100 UMIs for called SNPs) (Fig. 1c), we additionally tested Vireo with Vartrix-quantified SNPs (Vireo Vartrix). To enable the comparison of deconvolution methods Vireo CellSNP, Vireo Vartrix, and Souporcell, which cannot reassign cells to patients, with demultiplexing methods (i.e., Vireo Vartrix or CellSNP bulkRNA, Vireo Vartrix or CellSNP WES), we assigned deconvoluted cells to patients based on their overlap with reference clusters for Vireo CellSNP, Vireo Vartrix and Souporcell. Comparing recall and precision revealed that Souporcell, independent of the underlying SNP-quantification method, outperformed other methods, while Vireo CellSNP and Vireo Vartrix showed a slightly lower precision and recall compared to Souporcell (Fig. 1d). Despite the smaller number of SNPs, Vireo performed slightly better with CellSNP in comparison to Vartrix. Vireo Integration of bulkRNA or WES data to

enable demultiplexing led to a marginal improvement in precision, but a distinct loss in cell recall, with bulkRNA- outperforming WES-assisted demultiplexing. Further dissecting bulkRNA and WES-assisted demultiplexing revealed that bulkRNA recovered more SNPs from scRNA-seq data (Fig. 1e), particularly in the 3' untranslated region (3' UTR), which is primarily mapped during 3' scRNA-seq. While CellPlex was outperformed by genomic demultiplexing methods (Fig. 1d), stratifying patient assignment by clusters revealed a robust assignment of single cells for all methods including CellPlex (Fig. 1f). Indeed, quantifying the precision of assigned single cells (excluding doublets) revealed an effectively equal precision of all methods (~98.5%) (Fig. 1g). The poorer precision of CellPlex is therefore explained by an over-estimation of doublets (Fig. 1f). To assess doublet identification of multiplexing methods in more detail we next assessed the shared doublet assignment for each method including manually and computationally assigned doublets (scDblFinder[9]) using computationally identified doublets by scDblFinder as a reference (Supplementary Fig. 1g). In line with our prior observation, Vireo CellSNP bulkRNA and WES showed

the lowest overlap with scDblFinder (Fig. 1f) as both methods under-assign doublets (Supplementary Fig. 1g). In contrast, Souporcell showed the highest overlap with both manually and computationally inferred doublets by scDblFinder, highlighting its robust doublet assignment. Summarizing, Souporcell outperformed other demultiplexing methods (Fig. 1d), but offers no solution for patient assignment. While Vireo can assign patients, the latter leads to a distinct loss in recall. Lastly, bulkRNA is better suited for recovery of scRNA-seq SNPs than WES due to better coverage of the 3' UTR (Fig. 1d–e).

## SoupLadle framework

Based on our results, we decided to develop a framework, *SoupLadle*, that enables multiplexing by assigning Souporcell-deconvoluted cells back to patients (Fig. 1h). Within our framework, robust sample demultiplexing is ensured by the following steps: (1) bulkRNA-seq of each patient to select patients with distinct SNP-profiles for pooling (Fig. 1i). (2) Pooled scRNA-seq of selected patients. (3) Cell assignment to SNP-profiles with *Souporcell*. (4) Re-assignment of assigned SNP-profiles to patients based on the similarity of SNP-profiles to bulkRNA-seq SNPs using a hamming distance matrix and Kuhn-Munkres-Algorithm for assignment (Fig. 1j). We reasoned that a two-step process of cell demultiplexing and re-assignment would have the critical advantage that all called SNPs in scRNA-seq data can be considered for the initial cell deconvolution, rather than considering only SNPs that are recovered in reference bulkRNA or WES data. We hypothesized that this would be particularly critical for demultiplexing of rare cell-types with distinct gene expression and in consequence distinct SNP-profiles, as these transcripts and SNPs would be less well covered in bulkRNA-seq data. Indeed, calculating recall and precision for PBMC, while continuously randomly downsampling the number of bulkRNA-SNPs available for demultiplexing, confirmed a significantly superior performance of SoupLadle to Vireo, independent of the underlying SNP-quantification method used for Vireo (CellSNP or Vartrix) (Fig. 1k). The difference in performance between SoupLadle and Vireo methods further increased when continuously downsampling the number of bulkRNA-SNPs available for demultiplexing, underlining the notion that SoupLadle functions more robustly than Vireo when fewer SNPs are available.

## Benchmarking of SoupLadle framework in different tissues

To next test the ability of SoupLadle in single nuclear RNA sequencing (snRNA-seq) of a complex solid organ, we isolated nuclei from snap-frozen heart tissue of five patients, labeled nuclei with patient-specific CellPlex-Oligos and Hashtag-antibodies, and subsequently sorted and pooled nuclei for snRNA-seq (Fig. 2a, Supplementary Fig. 2a). In addition, we performed bulkRNA and WES of heart tissue from each patient (Supplementary Fig. 2b, c). After quality control, clustering clearly distinguished the major cell types of the heart (Supplementary Fig. 2d–f). Similar to our findings in scRNA data, SNP recovery was higher with Vartrix in comparison to CellSNP (Supplementary Fig. 2g). Assessing cell-to-patient assignment, SoupLadle outperformed both cell-labeling and genomic demultiplexing methods, assigning nearly all cells to patients (Fig. 2b). In comparison, Vireo bulkRNA coupled with Vartrix-quantified SNPs was able to achieve a similar assignment, while both cell labeling strategies and standard Vireo CellSNP bulkRNA only assigned ~75% of all cells. In contrast to scRNA-seq data, demultiplexing using WES (Vireo CellSNP WES) led to poor cell assignment (~12.5%). Indeed, the previously observed differences in SNP recovery were even more pronounced in snRNA-seq, as bulkRNA strongly outperformed WES SNP recovery due to better mapping of introns and 3'-UTRs (Fig. 2c). This is in line with the notion that unspliced RNA is more abundant within nuclei leading to a higher fraction of mapped introns.

Assessing the overlap of patient-assigned cells as a percentage of all assigned cells to estimate cell assignment quality revealed a high overlap (>95%) between all methods, with the exception of Vireo

CellSNP WES, confirming a robust cell assignment by SoupLadle, but also Cellplex, Hashtag and Vireo CellSNP or Vartrix bulkRNA (Fig. 2d, Supplementary Fig. 2h). To estimate cell-type bias, we last stratified cell assignment by celltype. Here, Hashtag cell-labeling, Vireo CellSNP bulkRNA and, to a lesser extent, Vireo Vartrix bulkRNA showed a distinct cardiomyocyte bias with poor assignment of less abundant cell populations (Fig. 2e). In contrast, CellPlex and SoupLadle showed a lower cell type bias (Fig. 2e).

Next, we decided to benchmark SoupLadle in a larger snRNA-seq dataset (*n* = 8, 11,501 nuclei) isolated from frozen human heart tissue (Fig. 2f, Supplementary Fig. 2i–k). Of note, due to excessive nuclei loss with hashtag or CMO labeling as a consequence of the required processing and washing steps, we did not perform cell labeling. Importantly, as both SoupLadle and Vireo outperformed cell labeling approaches in our previous benchmark (Fig. 1d), the latter cannot be considered a viable benchmark for SNP multiplexing tools. In line with our previous in-silico analysis (Fig. 1k), SoupLadle showed a robust performance with higher sample numbers, assigning 99% of cells to patients. In contrast, Vireo Vartrix bulkRNA cell assignment dropped with higher sample numbers to 81% assigned cells, while Vireo CellSNP bulkRNA only assigned 45% of cells (Fig. 2g). Assessing the overlap in patient assigned cells across multiplexing methods showed a 100% overlap between SoupLadle and Vireo Vartrix bulkRNA and a 96% overlap of SoupLadle and Vireo CellSNP bulkRNA, corroborating a robust cell assignment by SoupLadle and Vireo (Supplementary Fig. 2l, m). In line with our hypothesis that consideration of all SNPs in snRNA data is critical for the successful assignment of less abundant cell-types, SoupLadle showed less cell-type bias, with better demultiplexing of rare cell-types (e.g., Neuronal cluster) in comparison to both Vireo Vartrix bulkRNA and Vireo CellSNP bulkRNA (Fig. 2h).

To further validate our approach in (1) tissue characterized by a high cell heterogeneity and (2) smaller tissue samples, we next tested multiplexing of frozen human kidney biopsies (*n* = 4, 11,490 nuclei) (Fig. 2i, Supplementary Fig. 2n–p). Supporting our previous results, SoupLadle showed the highest cell assignment with 96% of cells assigned, followed by a drop in Vireo Vartrix bulkRNA demultiplexing performance with only 21% of cells assigned, while Vireo CellSNP bulkRNA failed to adequately demultiplex samples (<1% assigned) (Supplementary Fig. 2q). Again, overlap of assigned cells was high (-93%) in SoupLadle and Vireo Vartrix, while Vireo CellSNP did not adequately assign cells in comparison to other methods (Supplementary Fig. 2r, s). Analysis of kidney tissue strongly highlighted a Vireo Vartrix celltype bias, with inefficient demultiplexing of rare cell-types with distinct transcript expression such as podocytes, fibroblasts, and intercalated cells, in comparison to tubular cells (PT, TAL, and DCT), which represent the most common cell-type in the kidney (Fig. 2j). In contrast, SoupLadle showed no apparent cell-type bias, highlighting the advantage of our multiplexing approach (Fig. 2j).

Lastly, as we analyzed SoupLadle multiplexing performance in 3' scRNA-seq datasets only, we aimed to assess whether SoupLadle could be suitable for 5' scRNA-seq multiplexing. Here we analyzed SNP distribution in a published 5' scRNA-seq dataset (Supplementary Fig. 3a)[10]. In line with our 3' scRNA-seq analyses, we recovered a comparable amount of SNPs and found that the majority of recovered SNPs were located within introns, as well as 5' and 3' UTRs (Supplementary Fig. 3a, b, Fig. 1c, Supplementary Fig. 2g). These results suggest that bulkRNA-seq would be best suitable for 5' scRNA-seq multiplexing, as the latter better captures SNPs located within introns and UTRs than WES.

## Discussion

Comparing our approach to standard scRNA-seq costs, SoupLadle provides a -4-fold cost reduction when multiplexing eight samples (Supplementary Data 1). More importantly, one of the key advantages of SoupLadle in comparison to standard cell-labeling approaches

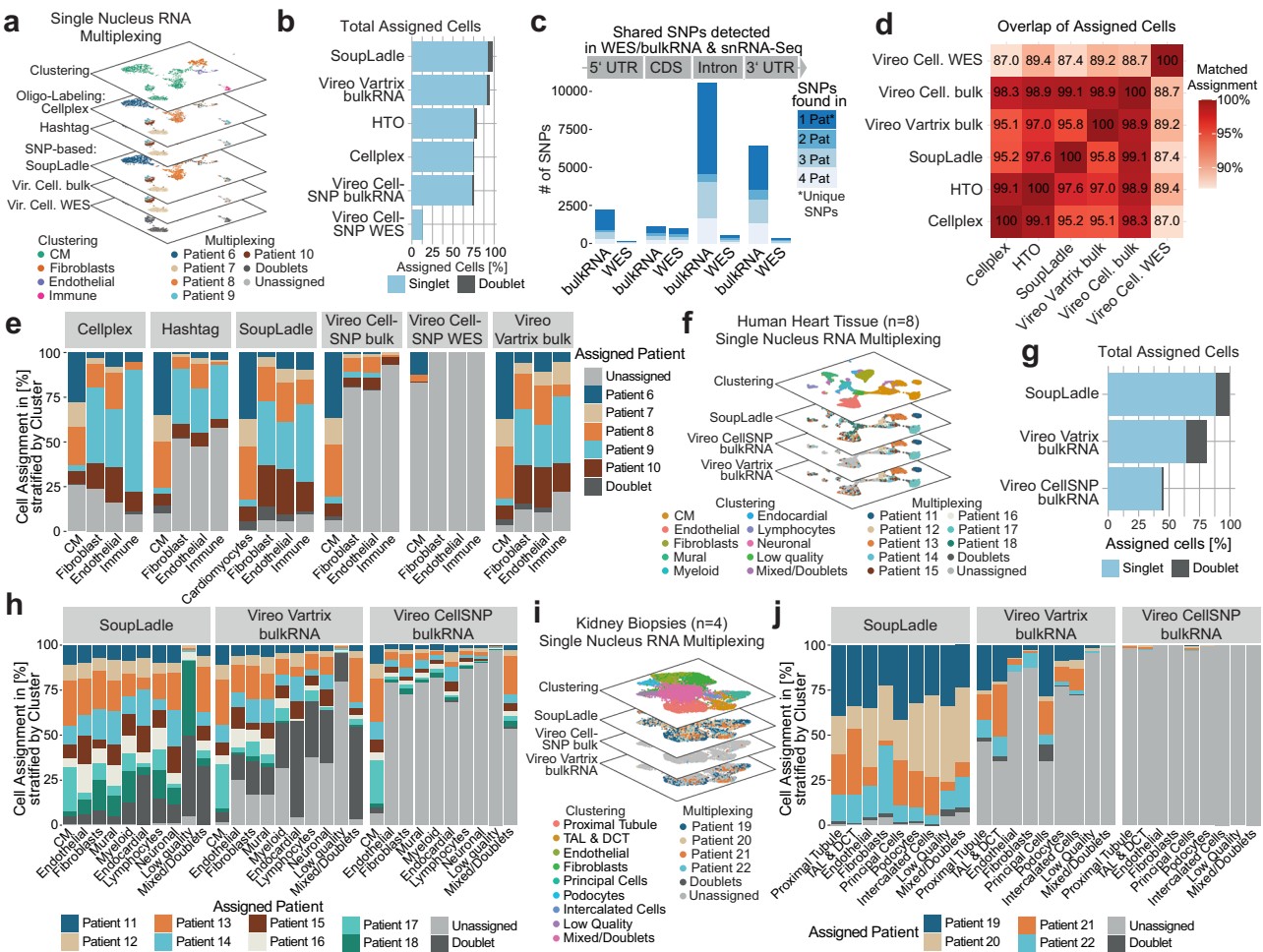

**Fig. 2 | Benchmarking of the SoupLadle multiplexing workflow across different datasets. a** Conceptual cartoon showing UMAP representations of cardiac nuclei (dataset 1, n = 5) stratified by clustering or patient assignment of multiplexing methods. Vir Vireo, bulk bulkRNA, Cell. CellSNP, CM Cardiomyocytes. **b** Cell assignment (patient-assigned singlets and doublets) in % of all cells. **c** Quantification of snRNA-seq SNPs recovered with bulkRNA or WES, split by genomic region. CDS coding sequence, UTR untranslated region. **d** Overlap of cell assignment for multiplexing methods in % of assigned cells. **e** Cell-to-patient assignment for each method stratified by celltype. CM Cardiomyocytes.

**f** Conceptual cartoon showing UMAP representations of cardiac nuclei (dataset 2, n = 8) stratified by clustering or patient assignment of multiplexing methods. **g** Total cell assignment of dataset 2 (patient-assigned singlets and doublets) in % of all cells. **h** Cell-to-patient assignment (dataset 2) for each method stratified by celltype. **i** Conceptual cartoon showing UMAP representations of kidney nuclei (dataset 3, n = 4) stratified by clustering or patient assignment of multiplexing methods. **j** Cell-to-patient assignment (dataset 3) for each method stratified by celltype. Exact numbers for cell sorting, and cell demultiplexing stratified by patient for each multiplexing method are provided in Supplementary Data 2.

(Hashtagging, Cellplex) is that it does not require additional experimental steps during cell or nuclei isolation. Instead, samples can be pooled immediately after cell or nuclei isolation, without having to label and wash samples prior to pooling. This reduces processing steps, which is critical for isolation of nuclei, as the latter are particularly sensitive to disruption or nuclei loss during washing steps (~20–30% of nuclei are lost with each washing step)[11]. In contrast to standard Cell-Labeling approaches, SoupLadle requires bulkRNA-seq or WES. Here, our data indicates that bulkRNA outperforms WES for SNP-based multiplexing, as WES does not capture 3′ UTR or intronic SNPs well. This is particularly important for snRNA-seq, where the majority of snRNA-seq SNPs are located in the 3′ UTR and introns. Taking into consideration that snRNA, but not scRNA, can be performed from frozen tissue, which is crucial for multiplexing large human cohorts, we recommend bulkRNA-seq for SNP-based multiplexing. While theoretically bulkRNA-seq can be performed in parallel to cell or nuclei isolation, we recommend an a priori evaluation of SNP heterogeneity (Fig. 1h). This additional step enables optimal selection of samples for pooling based on bulkRNA SNPs and avoids critical loss of samples in multiplexed scRNA/snRNA-data due to insufficient SNP

heterogeneity. As such, SoupLadle is best suited for frozen tissue, where bulkRNA-seq can be performed prior to multiplexing from a small piece (e.g., 1 mg) of frozen tissue. Of note, while we do not provide a tool for ambient RNA correction, ambient RNA can be imputed with standard packages (CellBender, SoupX) or Souporcell, which leverages the natural genetic variation of multiplexed patients to estimate ambient RNA.

In summary, SoupLadle enables robust multiplexing with higher recall as compared to other multiplexing methods. In comparison to standard scRNA-seq, our framework reduces costs and experimental batch effects without complicating the delicate workflow of cell or nuclei isolation. While we only provide a proof of concept for our framework, we believe that this multiplexing approach will enable multiplexing of larger sample numbers and will likely be instrumental to the generation of large-scale human cohorts.

## Methods
### Ethics
The use of PBMC for our purposes was approved by the scientific management of the RWTH centralized Biomaterial Bank (cBMB) and

the local Ethics Committee of the Medical Faculty of Medicine of the RWTH Aachen University. Human myocardial tissue was collected from patients undergoing heart transplantation, implantation of a total artificial heart, or left ventricular assist device implantation. The use of heart tissue was approved by the local ethics committee of the Ruhr University Bochum in Bad Oeynhausen (No. 220–640). The use of human kidney tissue was approved by the Medical Ethics Committee of the Erasmus Medical Center, Rotterdam (MEC-2021-0840). All patients provided written informed consent in accordance with the Declaration of Helsinki.

## PBMC isolation

For PBMC isolation, blood was collected from five patients (2 male, 3 female) into EDTA-tubes and mixed 1:1 with PBS. EDTA-blood was then carefully layered onto Ficoll-Paque Plus Cytiva (17-440-02, GE Healthcare) (1:1,5) and centrifuged (400 G, 40 min, RT, without brakes). After centrifugation the PBMC-Layer was aspirated, resuspended in FACS-Buffer (PBS, 2% FCS, 2 mM EDTA, Invitrogen, AM260G) and centrifuged (300 G, 10 min, RT). The supernatant was discarded and cells were resuspended in 300 µl FACS Buffer. At this step, PBMC were counted and ~0.5 × $10^6$ cells were taken for each, bulkRNA sequencing and WES. The remaining cells were stained with CD14-PE (1:100, Clone M5E2, 301850, Biolegend), CD16-APC (1:100, Clone B73.1, 360705, Biolegend), CD4-FITC (1:100, Clone RPA-T4, 300506, Biolegend), CD8-PE/Cy7 (1:100, Clone SK1, 344712, Biolegend) and CD19-BB700 (1:100, Clone SJ25C1, 566396, BD Biosciences) for 30 min at 4 °C, protected from light. Afterwards, cells were washed once with FACS Buffer (400 G, 5 min, 4 °C) before staining PBMC with a unique CellPlex-Oligo (3' CellPlex, 1000261, 10X) for each patient (100 µl, 5 min, RT). Subsequently cells were washed twice, before sorting a unique PBMC population for each patient (CD14 Monocytes, CD16 Monocytes, CD4 T-Cells, CD8 T-Cells, CD19 B-Cells) using a BD FACSMelody Cell Sorter. For lymphocytes Dapi, CD16 and CD14 positive cells were excluded to avoid dead cell and monocyte contamination, with subsequent sorting of CD4-CD8 + T-Cells, CD4 + CD8- T-Cells or CD4-CD8-CD19 + B-Cells (Supplementary Fig. 1a). For monocytes Dapi, CD4 and CD8 positive cells were excluded to avoid dead cell and lymphocyte contamination, with subsequent sorting of CD14 + CD16- or CD16 + CD14- monocytes (Supplementary Fig. 1a). For each sample we sorted an equal number of cells (100,000 cells per sample). After sorting, PBMC were pooled and immediately loaded onto a Chromium Next GEM Chip G for snRNA-seq (3' v3.1, 10X) with a cell recovery of ~18,000 cells after Cell Ranger alignment.

## Nuclei isolation from snap-frozen tissue

Snap frozen tissue was crushed using a mortar and pestle, resuspended in 500 µl nuclei lysis Buffer (EZ lysis Buffer, NUC101, Sigma-Aldrich with 1 Tab/10 ml of cOmplete Protease Inhibitor 11873580001, Roche and 10 µl/ml Recombinant RNase Inhibitor, 2313A, Takara Bio and 10 µl/ml Superase In RNase Inhibitor, AM2694, Thermofisher) and homogenized with dounce tissue grinder pestles. The homogenized solution was spun down, supernatant discarded, and the pellet resuspended in 4 ml nuclei resuspension buffer (PBS, 1% BSA, 126615-25 ML, Sigma-Aldrich, and 10 µl/ml Protector RNase Inhibitor, 3335399001, Roche, abbreviation: NRB). The suspension was then filtered via a 40 µm cell strainer and centrifuged (500 G, 4 °C, 5 min). Supernatant was discarded, and cells resuspended in 200 µl NRB with 3 µl of a unique TotalSeq anti-Nuclear Pore Complex Antibody Hashtag-Antibody (TotalSeq™ A0451-A0455, Biolegend) for each sample (4 °C, 20 min). Subsequently samples were washed with 1 ml NRB, centrifuged (500 G, 4 °C, 5 min) and resuspended in 100 µl of a unique CellPlex Oligo (3' CellPlex, 1000261, 10X) for each sample (RT, 5 min). The above steps for Hashtagging and CMO-Labeling were not performed for the larger heart dataset, where eight samples were pooled,

and the kidney biopsy dataset, as the additional processing steps led to excessive nuclei loss with insufficient remaining nuclei remaining for adequate chip loading (loading > 10,000 cells on the chip). Cells were washed once more with NRB (500 G, 4 °C, 5 min) before proceeding to Fluorescent-activated Nuclei-sorting of DAPI positive nuclei with a Sony SH800S. For the first heart dataset (n = 5; 2 male, 3 female) complete samples were sorted due to a low amount of recovered nuclei due to the additional required processing steps for Cellplex and Hashtag labeling (exact numbers provided in Supplementary Data 2), while for the second heart dataset an approximately equal number of nuclei per sample was sorted (~50,000 nuclei per patient) and subsequently pooled. For kidney biopsies (n = 4; 3 male, 1 female), where tissue was scarce, total processed biopsies were pooled prior to sorting to reduce sample loss. 5 min prior to sorting, nuclei were stained with DAPI (Sigma-Aldrich). After sorting, nuclei were pooled and immediately loaded onto a Chromium Next GEM Chip G for snRNA-seq (10x, 3' v3.1). The second heart cohort (n = 8; 5 male, 3 female) was loaded onto a Chromium Next GEM Chip M for High Throughput snRNA-seq (10x, 3' v3.1).

## Single cell RNA, CellPlex and Hashtag library preparation

Single cell RNA (3' v3.1, Dual Index, 10X for PBMC, heart cohort 1 and kidney biopsies; 3' High Throughput v3.1, Dual Index, 10X for heart cohort 2) and 3' CellPlex (3' CellPlex, 1000261, 10X) library preparation was performed according to the manufacturer's instructions. Hashtag-Libraries were prepared as described by Stoeckius et al.[3]. After quality control on an Agilent TapeStation, scRNA/snRNA-seq samples were sequenced on an Illumina NovaSeq system targeting a sequencing depth of 25000 reads/cell for scRNA and snRNA libraries. For CellPlex and Hashtag libraries we targeted a sequencing depth of 2500 reads per cell based on recommendations for sequencing depth of Hashtag/Cite-seq libraries (https://cite-seq.com/).

## BulkRNA library preparation, alignment, and SNP-calling

For snap frozen tissue samples, samples were shredded in RNeasy lysis buffer in a Mixer Mill prior to RNA Isolation. For PBMC, samples were lysed with RNeasy lysis buffer. Subsequently, RNA was extracted for both tissue and PBMC samples using the RNeasy Mini Kit (74106, Qiagen) according to the manufacturer's instructions. For PBMC and heart tissue, RNA libraries were prepped with the NEBNext Ultra II Directional RNA Library Prep Kit (NEB, E7760L) coupled with the NEBnext rRNA Depletion Kit (NEB E6310X) according to the manufacturer's instructions. For kidney biopsies the NEBNext Ultra II Directional RNA Library Prep Kit was performed without rRNA-depletion according to the manufacturer's instruction, to reduce costs, but also account for the small amount of input RNA available due to tissue scarcity. After quality control (Agilent TapeStation), bulkRNA-seq samples were sequenced on an Illumina NovaSeq system targeting 25 million reads/bulkRNA library. Sequencing reads from bulkRNA-seq were aligned with STAR to the same reference genome used for the single-cell RNA-seq and variants were called with freebayes using the following parameters: -iXu --min-mapping-quality 30 --min-base-quality 10 --min-coverage 5. The variant data was further processed with bcftools and R.

## Whole Exome Sequencing, alignment, and SNP-calling

Enrichment for the whole exome was performed using the Lotus™ DNA Library preparation kit (IDT, Coralville, Iowa, USA) according to the manufacturer's protocol. Sequencing was performed on a Nova-Seq6000 Sequencer (Illumina, San Diego, CA). FastQ-files were generated with bcl2fastq2 (Illumina). Sequencing reads from WES were aligned with bwa-mem (version 0.7.8) and variants were called with freebayes (version 1.3.6) or GATK (version 2.3.9) and further processed with bcftools and R.

## Single-cell RNA-seq demultiplexing

All absolute numbers for cell assignment are provided in Supplementary Data 2.

## Sample labeling

For demultiplexing of CellPlex and Hashtag based sample labeling, sequencing reads were mapped to the reference genome GRCh38-2020-A applying the cellranger multi pipeline (version 7.1.0) with default settings and the used sample barcodes for the corresponding tagging method.

## CellRanger alignment and scRNA-seq analysis

For scRNA-/snRNA-seq alignment CellRanger was run with standard settings (CellRanger version 7.1.0 and reference genome GRCh38-2020-A) for cell identification, but otherwise no QC cutoffs were set prior to demultiplexing, as the latter may confound demultiplexing analysis. Samples were subsequently analyzed with *Seurat*. First, samples were normalized and scaled (*NormalizeData*, *ScaleData*). For clustering we used 4000 highly variable genes to compute principal component analysis, and subsequent UMAP dimensionality reduction. Clustering was performed at the lowest resolution (*FindClusters*, res. 0.1) to identify major clusters.

## Vireo[4]

For the Vireo approach, cellSNP-lite[7] (Version 0.3.2) with the recommended setting of --minMAF 0.1 --minCOUNT 100 was used to call SNPs from the multiplexed aligned bam file. This setting defines SNP loci with at least 100 UMIs and 10% minor allele frequency. Furthermore the SNP calling was limited to the cell barcodes that were assigned as cells by the cell calling algorithm of the cell ranger multi pipeline. Subsequently, Vireo (version 0.5.7) was applied using the cellSNP-lite called variants and the number of donors mixed (-N 4-8). For the sample assignment with matching bulkRNA-seq or WES from donors, the overlapping SNPs with cellSNP-lite called variants were used as prior (-d --forceLearnGT). The same Vireo configuration was used for the Vireo Vartrix approach, with SNPs derived from Vartrix SNP calling as input.

## Souporcell[5]

For the Souporcell approach, candidate SNPs were identified using *freebayes*[8] (Github: https://github.com/freebayes/freebayes, version v1.3.6) from the multiplexed aligned bam file with the following parameters: --iXu --C 2 --q 20 --n 3 --E 1 --m 30 --min-coverage 6 --max-coverage 100000 --pooled-continuous. The called SNPs were subjected to allele counting with Vartrix (Github: https://github.com/10XGenomics/Vartrix; version 1.1.22) with parameters --umi--mapq 30 --scoring-method coverage. The resulting SNP count matrix, number of multiplexed samples (--num_clusters 4-8) as well as the cell barcodes that were assigned as cells by the cell calling algorithm by the cell ranger pipeline were subjected to cell clustering by genotype with Souporcell (version 2.4) and followed by doublet detection with troublet. For the Souporcell minimap approach, the multiplexed aligned bam file was remapped with minimap2 according to the authors recommendations before variant calling.

## SoupLadle for cohort SNP evaluation and patient reassignment

After SNP calling from patient samples (bulkRNA-seq or WES), SoupLadle provides an R package for processing and analysis of SNP profiles from the cohort, including evaluation of the SNP quality of each sample and identification of common and discriminatory SNPs that might support later single cell demultiplexing. This enables a selection of samples with the highest SNP diversity for optimal pooling. Soupladle also enables the curation of meta data that will be attached to the single-nuclei RNA-seq data. In the second part, the SNP profiles of deconvoluted single cells from scRNA-seq data will be matched to patient SNP profiles of analyzed bulkRNA data. As input, VCF files from (patient) bulkRNA SNPs and deconvoluted single-cell SNPs are required so that SoupLadle will work with any common single-cell RNA seq SNP-multiplexing method, although we recommend using Souporcell. The input VCF files will be processed, converted into a hamming distance matrix and assigned to the matching bulk sample by the Kuhn-Munkres-Algorithm. Alternatively, the deconvoluted patients can be assigned to a known group (e.g., Cluster, Cell tags). Finally, the assigned patients and metadata can be added to any common single-cell object (e.g., Seurat, AnnData), optionally in combination with the SNP profile as a separate assay.

## Doublet estimation

For doublet estimation from single-cell RNA-seq data, we applied *scDblFinder*[9], which simulates artificial doublets by computationally merging transcriptomes from pairs of randomly selected cells. We computed the doublet score for the top 4000 genes and determined cells with a score >3 as doublets. Subsequently, the overlapping assignments with all applied methods and manual cluster annotation were evaluated using scDblFinder-identified doublets as a reference.

## Precision-recall and SNP evaluation for patient assignments

True positives (TP) for each method were determined by counting the cells where cell assignment matched the correct reference clustering labels. False positives (FP) were calculated as the count of assigned cells of each method that matched any of the reference clustering labels, minus the TP. False negatives (FN) were calculated as the cell count of reference clustering labels minus the TP. Finally, precision was calculated as Precision=TP/(FP + TP) and Recall = TP/(FN + TP). For the performance evaluation of SoupLadle compared to Vireo CellSNP and Vireo Vartrix, the called SNPs from bulkRNA-seq of the PBMC dataset were 10 times randomly subsampled for different fractions from 1.0 (all SNPs used for demultiplexing) to 0.2 (downsampling to 20% of all bulkRNA-seq SNPs). For each fraction and method, the precision and recall as well as the number of assigned cells (including patient-assigned cells and doublets) were estimated. The different methods of each fraction were compared by a Wilcoxon signed-rank test.

## Reporting summary

Further information on research design is available in the Nature Portfolio Reporting Summary linked to this article.

## Data availability

Bulk- and scRNA-sequencing data generated in this study have been deposited in the GEO database under accession code GSE247708. Whole exome sequencing data have been deposited in the database of The European Genome-phenome Archive (EGA) with the accession code EGAD50000000928 and are available under restricted access for the protection of patient privacy. Access may be granted to qualified researchers for health/medical/biomedical purposes, who are bound by a Data Use Certification Agreement. Source data are provided as a source data file. Source data are provided with this paper.

## Code availability

All original code has been deposited at github and is available under the following link: https://github.com/ToreBle/SoupLadle. An archived version of the repository is also available on Zenodo [https://doi.org/10.5281/zenodo.13711299].

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

## Acknowledgements
This work was funded by RWTH Aachen University Clinician Scientist grants to K.H. and D.S., and a RWTH Aachen University START grant (139/21) to K.H. The work was also supported by CRU344, and RWTH START grant to S.H. This work was further supported by grants of the German Research Foundation (DFG: SFBTRR219 322900939, CRU344-4288578857858, CRU5011- 445703531), by a grant of the European Research Council (ERC-COG 101043403), a grant of the Else Kroener Fresenius Foundation (EKFS), the Dutch Kidney Foundation (DKF), TASKFORCE EP1805, by the BMBF eMed Consortia Fibromap, BMBF consortia CureFib and the ERA-CVD MENDAGE consortium (BMBF 01KL1907), the NWO VIDI 09150172010072 all to R.K. T.S. is supported by an excellence scholarship of the Else-Kröner-Fresenius Stiftung (2022_EKFS.03). The authors thank the entire team of the RWTH centralized Biomaterial Bank (RWTH cBMB).

## Author contributions
R.K., K.H., T.B., S.H., and D.S. conceived the idea and designed the study. K.H. and T.B. wrote the manuscript. K.H., D.S., A.B., and T.S. carried out wet-lab experiments with assistance from Q.L., L.Z., and C.M. R.M. and I.K. performed WES. T.B. performed data analysis with support of H.K., S.H., and K.H. H.M. organized cardiac tissue collection and biobanking. M.C. supported in kidney tissue collection. All authors contributed to editing the manuscript.

## Funding

## Competing interests
R.K. is a founder, shareholder and board member of Sequantrix GmbH, a member of the scientific advisory board of Hybridize Therapeutics, and has received honoraria for advisory boards and talks from Bayer, Chugai, Pfizer, Roche, Genentech, Lilly, and GSK and has received research funding from Travere Therapeutics, Galapagos, Novo Nordisk and Ask Bio. S.H. reports funding from Novo Nordisk and Ask Bio. K.H. and S.H. are co-founders of Sequantrix GmbH. The remaining authors declare no competing interests.
