## [Peer Review File · Nature Communications]

REVIEWER COMMENTS

Reviewer #1 (Remarks to the Author):

In this manuscript, Hoeft and colleagues benchmark various methods for single-cell RNA-sequencing deconvolution and demultiplexing. Two experiments are performed using samples from different donors, one using PBMCs and a second using cardiac cells of various lineages. The authors extend their analysis by developing and benchmarking a novel demultiplexing method (SoupLadle).

The manuscript begins with an overview of methods for deconvolution and demultiplexing. The authors motivate scRNA-seq multiplexing to mitigate the high costs of large sample sizes. They describe cell-labeling approaches (CellPlex / HashTagging), deconvolution using SNP based approaches (Vireo / Souporcell), and demultiplexing using patient SNPs from bulk RNA-seq or WES. The authors perform an initial experiment by labeling PBMCs from 5 patients with CellPlex, isolating a unique PBMC population from each via FACS, and deconvolving / demultiplexing using clustering on cell expression signatures as ground truth. They demonstrate that Souporcell achieves superior recall and precision but note that it does not perform demultiplexing. They also demonstrate that bulk RNA outperforms WES as a source of patient SNPs in this experiment.

The authors introduce SoupLadle, which performs deconvolution using Souporcell and assigns cells to patients using bulk RNA-derived patient SNPs and an assignment algorithm. The authors then perform multiplexed single nuclear RNA sequencing of 5 patient heart tissue samples to benchmark demultiplexing using all methods. The authors observe high recall with both SoupLadle and Vireo Vartrix bulk, however, SoupLadle significantly outperforms when downsampling the bulkRNA SNPs used for assignment. Overall, SoupLadle appears to demonstrate higher recall and lower bias than existing methods in this experiment.

The paper is clear and well written, although the experimental evidence of SoupLadle's performance and range of applications is somewhat limited.

Major issues/questions

- All benchmarked tools achieve high (98.5%) and comparable precision, and the claimed advantage of SoupLadle is improved recall by assigning more cells to individuals. However, the Vireo Vartrix bulkRNA method performs comparably in "Total Assigned Cells" (Fig 2.f) and cell-to-patient assignment (Fig 2.i). The variant downsampling results show the starkest difference in

performance between SoupLadle and Vireo Vartrix bulk (Fig 2.d). From this panel, it appears that selection of discriminatory SNPs between samples (performed by SoupLadle) is critical. The paper would be significantly strengthened by demonstrating the source of SoupLadle's claimed performance improvements. Is this driven primarily by selection of SNPs, the matching algorithm, or another factor?

- Both experiments are performed using relatively small sample sizes (5), and there is no discussion of how performance of these methods varies as a function of multiplexed samples. Perhaps SoupLadle could be applied to larger publicly available multiplexed scRNA-seq datasets that contain reference hashing information?

Minor issues/questions

- Why not evaluate and report SoupLadle performance on the first experiment?
- The Vartrix and CellSNP single cell RNA called SNPs (Fig. 1c) are vastly different. What accounts for this difference?
- The paper does differentiate between deconvolution and demultiplexing, but it is confusing to keep track of the nomenclature (e.g. Vireo Vartrix bulk vs Vireo bulkRNA). Is it possible to introduce more structured method names? (e.g. Vireo Vartrix bulkRNA, Vireo CellSNP bulkRNA).
- Fig. 1b: Value of these UMAP plots is not clear in this format and some plots are obscured.
- Fig. 1e: Is the conclusion that bulk RNA gets more intronic and 3' UTR SNPs than WES? Would that not be expected? Given the high utility of intronic information, what library prep workflow (polyA mRNA enrichment or rRNA depletion) was followed for the bulk RNAseq?
- Fig 2b: Panel is somewhat confusing and the light yellow color did not print well. The takeaway is the majority of SNPs are patient-specific? There might be a clearer way to highlight this.
- Related question: Does one need whole transcriptome/exon reference information to capture adequate SNPs for demultiplexing or do targeted sequencing panels (as used in oncology) provide a diverse enough set of SNPs?
- Fig. 2i: SoupLadle is claimed to have less cardiomyocyte bias than other methods including Hashtag. What is the likely source of this difference? Less efficient labeling of other cell types with Hashtags?
- Fig. 2e: UMAP obscured.
- A cartoon overview of the start-to-finish SoupLadle workflow (including labeling of each step with external packages/tools) would be helpful.

- A more thorough discussion of pros/cons/practical utility of SNP-based demultiplexing (which requires an additional reference bulk RNA sequence) versus sample Hashing would be informative for readers.

Reviewer #1 (Remarks on code availability):

Code is accessible at the Github link with README and examples from the manuscript, but I did not review in detail.

Reviewer #2 (Remarks to the Author):

The authors investigated and compared various multiplexing strategies for single cell RNAseq experiments. They developed an improved bioinformatic algorithm (SoupLadle) utilizing genetic polymorphisms (SNP's) to deconvolute pooled single cell suspensions. The authors clearly showed that the in silico use of transcriptome sequence SNP reads are superior compared to cell labelling techniques (oligo or antibody) for downstream sample discrimination.

Concerns:

The authors need to describe in more detail the advantages and disadvantages of using multiplexing strategies. Pooling equal numbers of PBMS from different individuals seems easier than e.g. mixing cells derived from a complex organ with multiple cell types and additional sampling variability. Can multiplexing be recommended for precious biopsy samples when doublet rates rise with increasing cell numbers submitted for scRNAseq on the droplet-based 10X Genomics platform?

What are the cost advantages when whole genome sequencing (\$500?) is required to generate enough SNP's for demultiplexing successfully? When pooling only 5 samples, cost advantages might not justify problems arising from pooling strategies.

The authors need to provide also absolute values for cells assigned to the specific sample origin not only percentages. How many cells from each sample has been submitted? How many cells

have been recovered? This will show whether equal numbers can be obtained from each sample or whether there are difficulties/variabilities with pooling even with PBMC's.

Which QC parameters (cutoffs) have been applied to identify the transcriptome for a cell? (How many genes per cell?). How many cells have been targeted (per run/ per sample)?

The authors showed that exome sequencing is insufficient to identify enough SNPs which can be used for the demultiplexing procedure after executing 3'RNAseq experiments. But many researchers use also 5' RNAseq kits. The authors may be able to calculate by using publicly available 5' data whether cheaper exome sequencing could be an alternative to WGS.

Ambient RNA is huge problem in scRNA transcriptics. There are bioinformatics tools to partially remove those in case of unlikely appearance in wrong cell types. When pooling various patient samples ambient RNA will show up in the wrong patient and might mislead downstream analysis and results and might be more difficult to remove with current tools. Have the authors thought about using their genetic markers to identify the amount of ambient RNA and how to remove those from the datasets?

Minor:

In online methods: please explain why 25,000 reads /cell have been used to sequence the scRNAseq libraries but only 2500 reads per cell for the CellPlex and Hashtag libraries! Is the poor performance of these label-free methods explainable with low sequence coverage?

In Main: Authors talk about "genetic multiplexing methods" and probably mixed it with "labelling methods" in this context because "genetic" implies the availability of SNP information.

In Methods: Probably a typo? usually 250 million sequence reads rather than 25 million sequence read depth used for sequencing each library (in case 10000 cells are targeted).

References: the following manuscript should be cited: "Multiplexed droplet single-cell RNA-sequencing using natural genetic variation", Hyun Min Kang et al., Nature Biotechnology volume 36, pages89–94 (2018).

Reviewer #2 (Remarks on code availability):

I am not a bioinformatician and not equipped to read code.

The authors would like to thank the reviewers for their expertise and careful assessment of our paper. We have taken all the comments into account and present to you a substantially modified and further improved manuscript, providing extensive new data in response to your comments.

Reviewer #1 (Remarks to the authors):

In this manuscript, Hoeft and colleagues benchmark various methods for single-cell RNA-sequencing deconvolution and demultiplexing. Two experiments are performed using samples from different donors, one using PBMCs and a second using cardiac cells of various lineages. The authors extend their analysis by developing and benchmarking a novel demultiplexing method (SoupLadle).

The manuscript begins with an overview of methods for deconvolution and demultiplexing. The authors motivate scRNA-seq multiplexing to mitigate the high costs of large sample sizes. They describe cell-labeling approaches (CellPlex / HashTagging), deconvolution using SNP based approaches (Vireo / Souporcell), and demultiplexing using patient SNPs from bulk RNA-seq or WES. The authors perform an initial experiment by labeling PBMCs from 5 patients with CellPlex, isolating a unique PBMC population from each via FACS, and deconvolving / demultiplexing using clustering on cell expression signatures as ground truth. They demonstrate that Souporcell achieves superior recall and precision but note that it does not perform demultiplexing. They also demonstrate that bulk RNA outperforms WES as a source of patient SNPs in this experiment.

The authors introduce SoupLadle, which performs deconvolution using Souporcell and assigns cells to patients using bulk RNA-derived patient SNPs and an assignment algorithm. The authors then perform multiplexed single nuclear RNA sequencing of 5 patient heart tissue samples to benchmark demultiplexing using all methods. The authors observe high recall with both SoupLadle and Vireo Vartrix bulk, however, SoupLadle significantly outperforms when downsampling the bulkRNA SNPs used for assignment. Overall, SoupLadle appears to demonstrate higher recall and lower bias than existing methods in this experiment.

The paper is clear and well written, although the experimental evidence of SoupLadle's performance and range of applications is somewhat limited.

Major issues/questions

1. All benchmarked tools achieve high (98.5%) and comparable precision, and the claimed advantage of SoupLadle is improved recall by assigning more cells to individuals. However, the Vireo Vartrix bulkRNA method performs comparably in "Total Assigned Cells" (Fig 2.f) and cell-to-patient assignment (Fig 2.i). The variant downsampling results show the starkest difference in performance between SoupLadle and Vireo Vartrix bulk (Fig 2.d). From this panel, it appears that selection of discriminatory SNPs between samples (performed by SoupLadle) is critical. The paper would be significantly strengthened by demonstrating the source of SoupLadle's claimed performance improvements. Is this driven primarily by selection of SNPs, the matching algorithm, or another factor?

The improvement in performance by SoupLadle is primarily driven by the cell to patient matching algorithm of SoupLadle in comparison to Vireo. SoupLadle first assigns (or deconvolutes) cells to distinct SNP profiles and subsequently assigns SNP profiles to patients

in a two step approach, while Vireo performs both simultaneously. Specifically, for SoupLadle, cells are first assigned to a SNP profile based on all available SNPs in scRNA/snRNA data (e.g. cell X is assigned to SNP-profile A, B, C, D or E, etc.). Second, the SNP profiles, to which cells were assigned, are matched to patients based on the best-matching overlap of deconvoluted SNP-profiles with SNPs recovered from reference bulkRNA/WES data. This approach considers all SNPs in scRNA/snRNA-seq data for the initial deconvolution, where recovered SNPs between different cell clusters (e.g. due to different transcripts being expressed) can vary strongly. Here, we hypothesize that the availability of as many SNPs as possible is critical for successful demultiplexing, and in particular critical for successful demultiplexing of rare cell-types with distinct gene expression, as the latter genes and their SNPs are likely not well covered in bulkRNA-seq data.

In contrast, Vireo demultiplexes scRNA cells based on SNPs recovered in bulkRNA data, and does not consider SNPs that were not recovered in bulkRNA data. The differences in performance are summarized in a **New Figure 1g**, which we have now revised by not only downsampling, but repeated random downsampling of SNPs. When a high amount of SNPs are available/recovered in bulkRNA-Seq, Vireo performance is significantly, but only slightly inferior to SoupLadle. However, with a decreasing number of SNPs available/recovered in bulkRNA-seq, Vireo demultiplexing performance drops, whereas SoupLadle shows a stable performance, as it considers all SNPs available within scRNA data. In line with the notion that this is particularly critical for rare cell types with distinct transcript expression, SoupLadle shows the least cell-type bias for rare cell-types such as immune cells, neuronal cells or podocytes in comparison to Vireo (**Figure 2e, h and j**).

Based on the comments of both Reviewer 1 and 2 we have additionally generated two novel datasets to assess SoupLadle and Vireo performance with (1) more patients pooled and (2) in a distinct, more complex organ, with low amount of available input tissue (kidney biopsies) (**New Figures 2f-j, Extended Data Fig. 2i-s**). First, we assessed SoupLadle and Vireo performance in a larger snRNA-seq dataset of frozen human heart tissue to validate successful demultiplexing of larger cohorts (n=8, 11501 nuclei). Second, we assessed demultiplexing performance in frozen kidney biopsies (n=4, 11490 nuclei), which are characterized by both a low amount of available tissue and a high degree of tissue-complexity, due to the high amount of distinct and also rare cell-types (Proximal Tubule, Distal Tubule, Fibroblasts, Endothelial Cells, Podocytes, Intercalated Cells, Principal Cells, and more). Of note, we did not perform either Hashtag or CMO labeling, as both approaches resulted in excessive nuclei loss due to the necessary washing steps and extended FACS time, which made (1) processing of larger sample numbers in parallel unfeasible and (2) led to too much sample loss for human kidney biopsy processing. Importantly, as both SoupLadle and Vireo outperformed cell-labeling strategies in terms of correct cell assignment to patients (**Figure 1d**), the latter cannot be considered as a reference benchmark for SNP-demultiplexing strategies.

Corroborating our hypothesis, we show that indeed with either a larger sample number (n=8) or small and complex tissue biopsies, Vireo Vartrix bulkRNA performance drops with only 81% (heart dataset) and 21% (kidney dataset) of cells assigned, while SoupLadle is still able to assign 99% (heart dataset) and 96% (kidney dataset) of cells (**New Figure 2g, New Extended Data Fig. 2q, New Supplemental Table 2**). As previously shown, Vireo Vartrix bulkRNA and SoupLadle still show a high overlap of assigned cells (100% for heart data,

92.9% for kidney data) corroborating the previously observed confidence in assignment (**New Extended Data Fig. 2l, s**). As expected, Vireo CellSNP bulkRNA drops even stronger with only 45% of cells assigned for the larger dataset, and fails to successfully demultiplex the kidney dataset with <1% of cells assigned and low overlap with Vireo Vartrix and SoupLadle (**New Figure 2g, New Extended Data Fig. 2l, q, s**). Lastly, as hypothesized, SoupLadle shows less cell-type bias and is able to demultiplex rare cell types, e.g. heart neuronal cells, kidney podocytes or intercalated cells, with a similar assignment rate as abundant-cell types (**New Figures 2h and j**). In contrast, both Vireo Vartrix bulkRNA and Vireo CellSNP bulkRNA show a cell-type bias for common cell-types (cardiomyocytes in hearts, proximal tubule and TAL & DCT in kidneys), which results in loss of rare cell-types (neuronal, lymphocytes, podocytes, intercalated cells) (**New Figures 2h and j**). Of note, we have adapted **Figure 2b** (previously **Figure 2f**), which previously showed total assigned singlets to patients in % of total cells, to now show both patient assigned cells/singlets and doublets in % of total recovered cells to be consistent with the new cell recovery figures (**New Figure 2f, Extended data 2q**). Exact cell assignments are also visible in **New Supplemental Table 2**.

To emphasize that the two step approach of first assigning cells to SNP-profiles based on all SNPs in scRNA-data, and then SNP-profiles to patients is the primary factor driving improved performance of SoupLadle, the authors have adapted the manuscript as follows. Changes to the manuscript are marked in gray.

“Based on our results we decided to develop a novel framework, SoupLadle, that enables multiplexing by assigning Souporecell-deconvoluted cells back to patients (Fig. 1h). Within our framework, robust sample demultiplexing is ensured by the following steps: (1) bulkRNA-seq of each patient to select patients with distinct SNP-profiles for pooling (Fig. 1i). (2) Pooled scRNA-seq of selected patients. (3) Cell assignment to SNP-profiles with Souporecell. (4) Re-assignment of assigned SNP-profiles to patients based on the similarity of SNP-profiles to bulkRNA-seq SNPs using a hamming distance matrix and Kuhn-Munkres-Algorithm for assignment (Fig. 1j). We reasoned that a two step process of cell demultiplexing and re-assignment would have the critical advantage that all called SNPs in scRNA-seq data can be considered for the initial cell deconvolution, rather than considering only SNPs that are recovered in reference bulkRNA or WES data. We hypothesized that this would be particularly critical for demultiplexing of rare cell-types with distinct gene expression and in consequence distinct SNP-profiles, as these transcripts and SNPs would be less well covered in bulkRNA-seq data. Indeed, calculating recall and precision for PBMC, while continuously randomly downsampling the number of bulkRNA-SNPs available for demultiplexing, confirmed a significantly superior performance of SoupLadle to Vireo, independent of the underlying SNP-quantification method used for Vireo (CellSNP or VarTriX) (Fig. 1k). The difference in performance between SoupLadle and Vireo methods further increased when continuously downsampling the number of bulkRNA-SNPs available for demultiplexing, underlining the notion that SoupLadle functions more robustly than Vireo when less SNPs are available.”

...

“Next, we decided to benchmark SoupLadle in a larger snRNA-seq dataset (n=8, 11501 nuclei) isolated from frozen human heart tissue (Figure 2f, Extended Data Fig. 2i-k). Of note, due to excessive nuclei loss with hashtag or CMO labeling as a consequence of the required processing and washing steps, we did not perform cell labeling. Importantly, as both

SoupLadle and Vireo outperformed cell labeling approaches in our previous benchmark (Figure 1d), the latter cannot be considered a viable benchmark for SNP multiplexing tools. In line with our previous in-silico analysis (Figure 1k), SoupLadle showed a robust performance with higher sample numbers, assigning 99% of cells to patients. In contrast, Vireo Vartrix bulkRNA cell assignment dropped with higher sample numbers to 81% assigned cells, while Vireo CellSNP bulkRNA only assigned 45% of cells (Figure 2g). Assessing the overlap in patient assigned cells across multiplexing methods showed a 100% overlap between SoupLadle and Vireo Vartrix bulkRNA and a 96% overlap of SoupLadle and Vireo CellSNP bulkRNA, corroborating a robust cell assignment by SoupLadle and Vireo (Extended Data Fig. 2l-m). In line with our hypothesis that consideration of all SNPs in snRNA data is critical for the successful assignment of less abundant cell-types, SoupLadle showed less cell-type bias, with better demultiplexing of rare cell-types (e.g. Neuronal cluster) in comparison to both Vireo Vartrix bulkRNA and Vireo CellSNP bulkRNA (Figure 2h).

To further validate our approach in (1) tissue characterized by a high cell heterogeneity and (2) smaller tissue samples, we next tested multiplexing of frozen human kidney biopsies (n=4, 11490 nuclei) (Figure 2i, Extended Data Fig. 2n-p). Supporting our previous results, SoupLadle showed the highest cell assignment with 96% of cells assigned, followed by a drop in Vireo Vartrix bulkRNA demultiplexing performance with only 21% of cells assigned, while Vireo CellSNP bulkRNA failed to adequately demultiplex samples (<1% assigned). Again, overlap of assigned cells was high (~93%) in SoupLadle and Vireo Vartrix, while Vireo CellSNP did not adequately assign cells in comparison to other methods (Extended Data Fig. 2r-s). Analysis of kidney tissue strongly highlighted a Vireo Vartrix celltype bias, with inefficient demultiplexing of rare cell-types with distinct transcript expression such as podocytes, fibroblasts, and intercalated cells, in comparison to tubular cells (PT, TAL & DCT), which represent the most common cell-type in the kidney (Figure 2j). In contrast, SoupLadle showed no apparent cell-type bias, highlighting the advantage of our novel multiplexing approach (Figure 2j).”

...

Online Methods

Nuclei isolation from snap-frozen tissue

“The above steps for Hashtagging and CMO-Labeling were not performed for the larger heart dataset, where eight samples were pooled, and the kidney biopsy dataset, as the additional processing steps led to excessive nuclei loss with insufficient remaining nuclei remaining for adequate chip loading (loading > 10000 cells on the chip).”

2. Both experiments are performed using relatively small sample sizes (5), and there is no discussion of how performance of these methods varies as a function of multiplexed samples. Perhaps SoupLadle could be applied to larger publicly available multiplexed scRNA-seq datasets that contain reference hashing information?

The authors agree with the reviewer and have now tested the performance of SNP multiplexing methods in a larger novel frozen heart snRNA-seq dataset, that is defined by a larger sample size (n=8) and cell number (n=11501), to test whether multiplexing performance changes with sample and cell number (New Figures 2f-j, Extended Data Fig. 2i-r). In

addition, we tested the performance of SNP multiplexing methods in frozen human kidney biopsies which are defined by a high tissue complexity and a low amount of input tissue available for nuclei isolation and bulkRNA-seq. As described above we not only show successful demultiplexing and less cell-type bias of SoupLadle, but confirm that, as hypothesized, Vireo performance drops with a larger sample size or higher tissue complexity, while SoupLadle performance is stable, assigning 99% of cells for heart tissue and 96% of cells for kidney biopsies (**New Figure 2f-j, Extended Data Fig. 2k-l, o-r**). The authors have adapted the manuscript as follows. Changes to the manuscript are marked in gray.

*“Next, we decided to benchmark SoupLadle in a larger snRNA-seq dataset (n=8, 11501 nuclei) isolated from frozen human heart tissue (**Figure 2f, Extended Data Fig. 2i-k**). Of note, due to excessive nuclei loss with hashtag or CMO labeling as a consequence of the required processing and washing steps, we did not perform cell labeling. Importantly, as both SoupLadle and Vireo outperformed cell labeling approaches in our previous benchmark (**Figure 1d**), the latter cannot be considered a viable benchmark for SNP multiplexing tools. In line with our previous in-silico analysis (**Figure 1k**), SoupLadle showed a robust performance with higher sample numbers, assigning 99% of cells to patients. In contrast, Vireo Vartrix bulkRNA cell assignment dropped with higher sample numbers to 81% assigned cells, while Vireo CellSNP bulkRNA only assigned 45% of cells (**Figure 2g**). Assessing the overlap in patient assigned cells across multiplexing methods showed a 100% overlap between SoupLadle and Vireo Vartrix bulkRNA and a 96% overlap of SoupLadle and Vireo CellSNP bulkRNA, corroborating a robust cell assignment by SoupLadle and Vireo (**Extended Data Fig. 2l-m**). In line with our hypothesis that consideration of all SNPs in snRNA data is critical for the successful assignment of less abundant cell-types, SoupLadle showed less cell-type bias, with better demultiplexing of rare cell-types (e.g. Neuronal cluster) in comparison to both Vireo Vartrix bulkRNA and Vireo CellSNP bulkRNA (**Figure 2h**).*

*To further validate our approach in (1) tissue characterized by a high cell heterogeneity and (2) smaller tissue samples, we next tested multiplexing of frozen human kidney biopsies (n=4, 11490 nuclei) (**Figure 2i, Extended Data Fig. 2n-p**). Supporting our previous results, SoupLadle showed the highest cell assignment with 96% of cells assigned, followed by a drop in Vireo Vartrix bulkRNA demultiplexing performance with only 21% of cells assigned, while Vireo CellSNP bulkRNA failed to adequately demultiplex samples (<1% assigned). Again, overlap of assigned cells was high (~93%) in SoupLadle and Vireo Vartrix, while Vireo CellSNP did not adequately assign cells in comparison to other methods (**Extended Data Fig. 2r-s**). Analysis of kidney tissue strongly highlighted a Vireo Vartrix celltype bias, with inefficient demultiplexing of rare cell-types with distinct transcript expression such as podocytes, fibroblasts, and intercalated cells, in comparison to tubular cells (PT, TAL & DCT), which represent the most common cell-type in the kidney (**Figure 2j**). In contrast, SoupLadle showed no apparent cell-type bias, highlighting the advantage of our novel multiplexing approach (**Figure 2j**).”*

Minor issues/questions

3. Why not evaluate and report SoupLadle performance on the first experiment?

For the first experiment using the sorted PBMC data we first benchmarked SoupLadle and assigned deconvoluted cells manually to patients based on overlap with sorted reference

clusters, due to the inability of SoupCell to assign deconvoluted cells to patients. Subsequently, we tested our novel SoupLadle approach in PBMC data by assigning deconvoluted cells computationally to matching bulkRNA-seq samples by the Kuhn-Munkres-Algorithm resulting in a reliable assignment of SNP-clusters to the correct patient (**Figure 1j**). Next, we evaluated cell to patient assignment and precision of SoupLadle in the PBMC dataset in comparison to Vireo cellSNP and Vireo Vartrix by random downsampling of bulkRNA-seq SNPs (**Figure 1k**). This demonstrated that SoupLadle robustly and precisely assigns the deconvoluted cells to patients.

4. The Vartrix and CellSNP single cell RNA called SNPs (Fig. 1c) are vastly different. What accounts for this difference?

The major differences in freebayes/Vartrix and CellSNP are explained by the different methods and parameters used, following the recommendations for SNP calling in single-cell RNA-seq data. While for cellSNP, SNP loci with at least 100 UMIs and 10% minor allele frequency are considered, the criteria for freebayes are more focused on the base quality at the SNP site. Here only SNPs with a base quality of 20, a minimum mapping quality of 30 and at least two observations supporting the alternate allele are considered. To exclude the possibility that Vireo simply performs worse due to fewer input SNPs for multiplexing, we also performed Vireo based on Vartrix-SNPs, showing that the improvement in performance of SoupCell and SoupLadle is not simply a consequence of different underlying SNP calling algorithms. We have adapted the manuscript as follows to reflect this. Changes are marked in gray.

“SNPs were quantified as recommended with CellSNP for Vireo and either VarTrix or Minimap for SoupCell (SoupCell or SoupCell Minimap). As VarTrix recovered considerably more SNPs than CellSNP due to stricter filtering of SNPs (i.e. minimum of 100 UMIs for called SNPs) (Fig. 1c), we additionally tested Vireo with VarTrix-quantified SNPs (Vireo Vartrix).”

5. The paper does differentiate between deconvolution and demultiplexing, but it is confusing to keep track of the nomenclature (e.g. Vireo Vartrix bulk vs Vireo bulkRNA). Is it possible to introduce more structured method names? (e.g. Vireo Vartrix bulkRNA, Vireo CellSNP bulkRNA).

As suggested by the reviewer, the authors have introduced more structured method names, labeling Vireo methods either as *Vireo Vartrix* or *Vireo CellSNP* depending on the underlying algorithm for SNP quantification. While the authors agree, that the nomenclature of deconvolution and demultiplexing can be confusing, the authors think it is important to emphasize the difference between (1) deconvolution, where cells in scRNA-data are assigned to different SNP profiles, but not re-assigned to the original patient, and (2) demultiplexing, where cells are assigned to patients based on reference SNP-data. However, the authors agree that this nomenclature makes it at times unnecessarily difficult to follow, and have therefore reduced it as much as possible throughout the manuscript, replacing it with specific method names, as suggested by the reviewer, when possible. The manuscript was adapted as follows, changes are marked in gray.

Main

“To enable the comparison of the deconvolution methods Vireo CellSNP, Vireo Vartrix and Souporcell, which cannot reassign cells to patients, with demultiplexing methods (i.e. Vireo Vartrix or CellSNP bulkRNA, Vireo Vartrix or CellSNP WES), we assigned deconvoluted cells to patients based on their overlap with reference clusters for Vireo CellSNP, Vireo Vartrix and Souporcell.”

Figure Legends:

Figure 1:

*...“j, Heatmap of normalized hamming distance for best-matching patient assignment with SoupLadle between SNPs from Souporcell-deconvoluted PBMC and of scRNA-seq PBMC SNP profiles to patient bulkRNA-SNP profiles. Deconvoluted cells SNP profiles were assigned to patients with the least normalized distance (labeled with #)... *Deconvolution methods Vireo CellSNP, Vireo Vartrix, and Souporcell were assigned to patients based on overlap with clustering.”*

Methods

“After SNP calling from patient samples (bulkRNA-seq or WES), SoupLadle provides an R package for processing and analysis of SNP profiles from the cohort, including evaluation of the SNP quality of each sample and identification of common and discriminatory SNPs that might support later single cell demultiplexing deconvolution. This enables a selection of samples with the highest SNP diversity for optimal pooling. Soupladle also enables the curation of meta data that will be attached to the single-nuclei RNA-seq data. In the second part, the SNP profiles of deconvoluted single cells from single-cell RNA sequencing data will be matched to patient SNP profiles of analyzed bulkRNA data. As input, VCF files from (patient) bulkRNA SNPs and deconvoluted single-cell SNPs are required so that SoupLadle will work with any common single-cell RNA seq deconvolution-SNP-multiplexing method, although we recommend using Souporcell...”

6. Fig. 1b: Value of these UMAP plots is not clear in this format and some plots are obscured.

The authors thank the reviewer for their observation, and have now reuploaded the Figures as separate high-resolution files (600 dpi).

7. Fig. 1e: Is the conclusion that bulk RNA gets more intronic and 3' UTR SNPs than WES? Would that not be expected? Given the high utility of intronic information, what library prep workflow (polyA mRNA enrichment or rRNA depletion) was followed for the bulk RNAseq?

For bulkRNA-seq we followed an rRNA depletion workflow for the first three datasets (PBMC, heart tissue datasets), which as seen in **Figure 1e** and **Figure 2g** captures both exonic and intronic SNPs. Of note, in **Figure 1e** and **2g** we subset exons into coding regions and untranslated regions (UTR) (see PMID: 37082142 for a detailed article about exonic regions), as the UTR also contain exons. As such we aim to emphasize that while the UTR contains exonic regions, WES does not map these efficiently, but instead only maps coding regions well. For kidney biopsies we performed the NEBNext Ultra II Directional RNA Library

Prep Kit without prior rRNA-depletion according to the manufacturer's instruction, to reduce costs, but also account for the small amount of input RNA available due to tissue scarcity. Here, we show that successful multiplexing is also possible without prior rRNA depletion, which reduces overall costs per sample, but also the amount of input RNA needed for bulkRNA library preparation. In addition, the authors want to emphasize with **Figure 1e** and **2g** the stark differences in SNP-distribution between scRNA-seq and snRNA-seq. As snRNA-seq contains more unspliced nuclear RNA, the latter is characterized by a much higher recovery of 3' UTR and intronic SNPs, while SNPs in the coding region, which is best mapped by WES, represent only a small fraction of all SNPs.

As snRNA-seq, but not scRNA-seq, can be performed from frozen tissue (in contrast to scRNA, which has to be performed on live cells), it is the most common experimental method for sequencing of stored human tissue samples. In contrast, scRNA-seq is only possible with live cells. Here, the interest for multiplexing approaches is not as high in our view, as human tissue samples for research are typically gained within medically-indicated interventions (e.g. kidney or cardiac biopsy for unclear organ failure), where multiplexing is simply not an option as typically multiple biopsies are not performed within the same day. In contrast, snRNA-seq can be performed from frozen tissue, which enables the collection and storage of samples over any time frame, and subsequent multiplexing of samples, when sufficient samples have been collected. Considering that (1) snRNA-seq is the preferred method for multiplexing as it can be performed on frozen samples and (2) bulkRNA-seq is in particular superior to WES for snRNA-seq, we aim to emphasize with the two panels why we choose bulkRNA-seq over WES for multiplexing. The manuscript was adapted as follows. Changes to the manuscript are marked in gray.

“Here, our data indicates that bulkRNA outperforms WES for SNP-based multiplexing, as WES does not capture 3' UTR or intronic SNPs well. This is particularly important for snRNA-seq, where the majority of snRNA-seq SNPs are located in the 3' UTR and introns. Taking into consideration that snRNA, but not scRNA, can be performed from frozen tissue, which is crucial for multiplexing large human cohorts, we recommend bulkRNA-seq for SNP-based multiplexing.”

Online Methods

“For PBMC and heart tissue RNA libraries were prepped with the NEBNext Ultra II Directional RNA Library Prep Kit (NEB, E7760L) coupled with the NEBnext rRNA Depletion Kit (NEB E6310X) according to the manufacturer's instructions. For kidney biopsies the NEBNext Ultra II Directional RNA Library Prep Kit was performed without rRNA-depletion according to the manufacturer's instruction, to reduce costs, but also account for the small amount of input RNA available due to tissue scarcity. ”

8. Fig 2b: Panel is somewhat confusing and the light yellow color did not print well. The takeaway is the majority of SNPs are patient-specific? There might be a clearer way to highlight this.

The authors agree with the reviewer and have now changed the color scheme of panel 2b (**New Figure 1i**). In Figure 1e we show that the majority of SNPs are patient-specific being present only in one patient. However, this does not reveal if the patient-unique SNPs are

distributed equally between patients. This is crucial for the deconvolution as in the case of two genetically similar patients demultiplexing of all patients might not be possible. The main concept that the authors therefore aim to convey here is that in the proposed *SoupLadle* workflow, we recommend to perform bulkRNA-sequencing of the to-be-sequenced patient cohort first, and only if a sufficient number of patient-unique SNPs is identified in bulkRNA, pool genetically distinct samples and perform scRNA or snRNA-multiplexing (e.g. here > 10000 unique SNPs per patient). This approach will enable sequencing of large cohorts (n>100, pooling n = 8 samples per run), while minimizing risk of unsuccessful demultiplexing due to two or more samples being genetically too similar.

9. Related question: Does one need whole transcriptome/exon reference information to capture adequate SNPs for demultiplexing or do targeted sequencing panels (as used in oncology) provide a diverse enough set of SNPs?

Indeed, the authors believe that a targeted genotyping array/panel of the 3' UTR and and/or intron regions could be sufficient for sample multiplexing. However, as SNPs recovered in scRNA/snRNA-seq data can vary considerably depending on the tissue or cell-type sequenced, a targeted SNP panel would likely have to differ depending on the tissue or cell-type of interest. Considering that (1) the costs of bulkRNA-seq are marginally higher and will likely continue to drop, (2) bulkRNA-seq provides additional information on patient samples, including more detailed information on SNPs and transcript expression and (3) the additional time needed for designing a specific SNP-panel for each tissue of interest, we believe that bulkRNA-Seq is the more robust choice for sample multiplexing.”

10. Fig. 2i: SoupLadle is claimed to have less cardiomyocyte bias than other methods including Hashtag. What is the likely source of this difference? Less efficient labeling of other cell types with Hashtags?

We hypothesize that the cause of the observed celltype bias differs depending on the underlying multiplexing method. For hashtag antibodies we suspect that hashtag anti-nuclear pore complex antibodies bind cardiomyocyte (CM) nuclei more efficiently due to different expression of nuclear pore complex (antigens) in nuclei of different cell types. Here we hypothesize that Cellplex shows less cell-type bias as the lipid-anchor tagged oligonucleotides integrate into cell and nuclei membranes in an unbiased manner as they do not rely on expression of target-antigens (**Figure 2e**).

In case of SNP demultiplexing algorithms we hypothesize that the lesser cell type bias of SoupLadle is explained by differences in demultiplexing approaches of Vireo and SoupORcell/SoupLadle as explained above in regards to Question 1. In short, SoupLadle first demultiplexes and assigns cells to SNP-profiles based on availability of all SNPs in scRNA-seq data, and then matches SNP profiles to patients based on overlap of SNP-profiles with bulkRNA data (e.g. SNP-Profile 1 to patient 3, SNP-Profile 2 to patient 5). As rare cell-types with distinct gene expression profiles and thus distinct SNP profiles are not well covered in bulkRNA sequencing data, it is likely that SNPs from these genes will be underrepresented in bulkRNA-seq data. Thus, algorithms such as Vireo, that only consider bulkRNA-called SNPs for demultiplexing will likely perform worse for rare cell types. In contrast, SoupLadle leverages

all identified SNPs in patient scRNA-/snRNA-seq data to deconvolute cells to SNP-profiles and then matches SNP-profiles to patients, which we hypothesize is the reason for the decreased cell-type bias. Indeed, both datasets that we created within the revision, (1) one of frozen human heart tissue with a larger sample number, and (2) a second of human kidney biopsies, a tissue defined by high cell heterogeneity, confirmed our hypothesis that SoupLadle shows no apparent cell-type bias (**New Figure 2h and j**). In contrast, Vireo Vartrix bulkRNA and Vireo CellSNP bulkRNA show a bias for abundant cell-types (cardiomyocytes in the heart, tubular cells in the kidney), with loss of rare cells such as neuronal cells, or podocytes (**New Figure 2h and j**). We have adapted the manuscript as follows to reflect these hypotheses. Changes to the manuscript are marked in gray.

“We reasoned that a two step process of cell demultiplexing and re-assignment would have the critical advantage that all called SNPs in scRNA-seq data can be considered for the initial cell deconvolution, rather than considering only SNPs that are recovered in reference bulkRNA or WES data. We hypothesized that this would be particularly critical for demultiplexing of rare cell-types with distinct gene expression and in consequence distinct SNP-profiles, as these transcripts and SNPs would be less well covered in bulkRNA-seq data.”

...

*“Next, we decided to benchmark SoupLadle in a larger snRNA-seq dataset (n=8, 11501 nuclei) isolated from frozen human heart tissue (**Figure 2f, Extended Data Fig. 2i-k**). Of note, due to excessive nuclei loss with hashtag or CMO labeling as a consequence of the required processing and washing steps, we did not perform cell labeling. Importantly, as both SoupLadle and Vireo outperformed cell labeling approaches in our previous benchmark (**Figure 1d**), the latter cannot be considered a viable benchmark for SNP multiplexing tools. In line with our previous in-silico analysis (**Figure 1k**), SoupLadle showed a robust performance with higher sample numbers, assigning 99% of cells to patients. In contrast, Vireo Vartrix bulkRNA cell assignment dropped with higher sample numbers to 81% assigned cells, while Vireo CellSNP bulkRNA only assigned 45% of cells (**Figure 2g**). Assessing the overlap in patient assigned cells across multiplexing methods showed a 100% overlap between SoupLadle and Vireo Vartrix bulkRNA and a 96% overlap of SoupLadle and Vireo CellSNP bulkRNA, corroborating a robust cell assignment by SoupLadle and Vireo (**Extended Data Fig. 2l-m**). In line with our hypothesis that consideration of all SNPs in snRNA data is critical for the successful assignment of less abundant cell-types, SoupLadle showed less cell-type bias, with better demultiplexing of rare cell-types (e.g. Neuronal cluster) in comparison to both Vireo Vartrix bulkRNA and Vireo CellSNP bulkRNA (**Figure 2h**).*

*To further validate our approach in (1) tissue characterized by a high cell heterogeneity and (2) smaller tissue samples, we next tested multiplexing of frozen human kidney biopsies (n=4, 11490 nuclei) (**Figure 2i, Extended Data Fig. 2n-p**). Supporting our previous results, SoupLadle showed the highest cell assignment with 96% of cells assigned, followed by a drop in Vireo Vartrix bulkRNA demultiplexing performance with only 21% of cells assigned, while Vireo CellSNP bulkRNA failed to adequately demultiplex samples (<1% assigned). Again, overlap of assigned cells was high (~93%) in SoupLadle and Vireo Vartrix, while Vireo CellSNP did not adequately assign cells in comparison to other methods (**Extended Data Fig. 2r-s**). Analysis of kidney tissue strongly highlighted a Vireo Vartrix celltype bias, with inefficient demultiplexing of rare cell-types with distinct transcript expression such as podocytes, fibroblasts, and intercalated cells, in comparison to tubular cells (PT, TAL & DCT), which*

represent the most common cell-type in the kidney (**Figure 2j**). In contrast, SoupLadle showed no apparent cell-type bias, highlighting the advantage of our novel multiplexing approach (**Figure 2j**).”

11. Fig. 2e: UMAP obscured.

The authors thank the reviewer for their observation, and have now reuploaded the Figures as separate high-resolution files (600 dpi).

12. A cartoon overview of the start-to-finish SoupLadle workflow (including labeling of each step with external packages/tools) would be helpful.

The authors agree with the reviewer and have now provided a cartoon overview for the SoupLadle workflow with labeling of external called packages/tools as requested by the reviewer (**New Figure 1h**).

13. A more thorough discussion of pros/cons/practical utility of SNP-based demultiplexing (which requires an additional reference bulk RNA sequence) versus sample Hashing would be informative for readers.

The authors now provide a more detailed discussion of pros/cons/practical utility of SNP-based demultiplexing including a novel table (**Supplemental Table 1**) that calculates cost-advantages for SoupLadle in comparison to standard single cell RNA sequencing (changes are marked in gray).

“Comparing our novel approach to standard scRNA-seq costs, SoupLadle provides a ~4-fold cost reduction (even higher when performing bulkRNA-seq without rRNA-depletion) when multiplexing eight samples (**Supplemental Table 1**). More importantly, one of the key advantages of SoupLadle in comparison to standard cell-labeling approaches (Hashtagging, Cellplex) is that it does not require additional experimental steps during cell isolation. Instead, samples can be pooled immediately after cell or nuclei isolation, without having to label and wash samples prior to pooling. This reduces processing steps, which is critical for isolation of nuclei, as the latter are particularly sensitive to disruption or nuclei loss during washing steps (~20-30% of nuclei are lost with each washing step)¹⁰. In contrast to standard Cell-Labeling approaches, SoupLadle requires bulkRNA-seq or WES. Here, our data indicates that bulkRNA outperforms WES for SNP-based multiplexing, as WES does not capture 3' UTR or intronic SNPs well. This is particularly important for snRNA-seq, where the majority of snRNA-seq SNPs are located in the 3' UTR and introns. Taking into consideration that snRNA, but not scRNA, can be performed from frozen tissue, which is crucial for multiplexing large human cohorts, we recommend bulkRNA-seq for SNP-based multiplexing. While theoretically bulkRNA-seq can be performed in parallel to cell or nuclei isolation, we recommend an a priori evaluation of SNP heterogeneity (**SoupLadle Workflow, Figure 1h**). This additional step enables optimal selection of samples for pooling based on bulkRNA SNPs and avoids critical loss of samples in multiplexed scRNA/snRNA-data due to insufficient SNP heterogeneity. As such, SoupLadle is best suited for frozen tissue, where bulkRNA-seq can be performed prior to multiplexing from a small piece (e.g. 1 mg) of frozen tissue.”

Reviewer #1 (Remarks on code availability):

Code is accessible at the Github link with README and examples from the manuscript, but I did not review in detail.

Reviewer #2 (Remarks to the authors):

The authors investigated and compared various multiplexing strategies for single cell RNAseq experiments. They developed an improved bioinformatic algorithm (SoupLadle) utilizing genetic polymorphisms (SNP's) to deconvolute pooled single cell suspensions. The authors clearly showed that the in silico use of transcriptome sequence SNP reads are superior compared to cell labelling techniques (oligo or antibody) for downstream sample discrimination.

Concerns:

1. The authors need to describe in more detail the advantages and disadvantages of using multiplexing strategies. Pooling equal numbers of PBMS from different individuals seems easier than e.g. mixing cells derived from a complex organ with multiple cell types and additional sampling variability. Can multiplexing be recommended for precious biopsy samples when doublet rates rise with increasing cell numbers submitted for scRNAseq on the droplet-based 10X Genomics platform?

The authors agree with the reviewer and have therefore performed two novel multiplexing experiments. We now demonstrate, as described in Response to Reviewer 1 Question 1 the feasibility of SoupLadle multiplexing with a larger number of patients (n=8) and nuclei (n=11501) in frozen human heart tissue (**New Figure 2f-h, New Extended Data Fig. 2i-m**). In a second dataset we validate SoupLadle multiplexing in a distinct organ by assessing multiplexing in frozen human kidney biopsies (n=4, 11490 nuclei), where tissue complexity is high, and the amount of tissue available for multiplexing is small (**New Figure 2i-j, New Extended Data Fig. 2n-s**). Of note, we did not perform Hashtag or CMO labeling for either experiment, as the required processing steps and extended FACS-Sorting led to excessive nuclei loss (also seen for heart dataset 1, where the number of sorted nuclei was low), making these multiplexing approaches infeasible for both larger sample numbers and kidney biopsies.

We now show that SoupLadle can robustly demultiplex samples in case of higher sample numbers (99% of cells assigned for heart dataset) and in tissues with higher celltype complexity (96% cells assigned for kidney dataset) (**New Figure 2g, New Extended Data Fig. 2q**). Importantly, we show that with larger sample numbers and more complex tissue types the differences in performance between SoupLadle and Vireo are amplified (**New Figure 2g-h, j, New Extended Data Fig. 2q**). Here, Vireo Vartrix bulkRNA and in particular Vireo CellSNP bulkRNA performance strongly dropped with higher cell numbers and tissue complexity with only 81% and 21% for Vireo Vartrix bulkRNA and 45% and <1% of cells assigned for Vireo CellSNP bulkRNA (**New Figure 2g-h, j, New Extended Data Fig. 2q**). Importantly, assessing the overlap of assigned cells of Vireo Vartrix and SoupLadle still showed a strong overlap (100% for heart dataset, ~93% for kidney dataset), corroborating the high similarity of assigned cells also observed in previous datasets (**New Extended Data Fig. 2l, s**). In summary, we here show that SoupLadle multiplexing is feasible and strongly outperforms Vireo for larger sample numbers, more complex tissue types, and smaller tissue samples.

As the reviewer points out, doublet detection is critical when multiplexing rare tissue samples. However, as others have shown previously, multiplexing approaches actually improve doublet detection, as doublets from different patients or samples can more reliably be identified based on the presence of patient-specific SNPs or Oligo-tags from two or more

patients within the same droplet (PMID: 30567574, 29227470). Similarly, the SoupLadle workflow detects/estimates doublets based on available SNP-information leveraging SoupCell. Here our initial benchmark study in PBMC indicated that SoupCell not only robustly identifies doublets, but outperforms Cellplex- and Vireo-based doublet identification (**Figure 1f**). Indeed, we now assessed the overlap of identified doublets for each method including the overlap with manually annotated doublets (based on clustering/DEG information) and computationally-inferred doublets by scDbtFinder (PMID: 35814628) using scDbtFinder doublets as a reference (**New Extended Data Fig. 1g**). Indeed, our overlap analysis confirmed that SoupCell had the highest overlap of all multiplexing methods with manually and computationally identified doublets in our PBMC dataset (**New Extended Data Fig. 1g**). In contrast, our novel analysis highlights the lower overlap of Vireo CellSNP WES and Vireo CellSNP bulkRNA identified doublets with both manually and computationally inferred doublets in line with our prior data suggesting that both methods underassign doublets (**Figure 1f**). Lastly, we recommend estimation of expected doublets prior to multiplexing with available online tools (e.g. <https://satijalab.org/costpercell/>). As such, the combination of estimation of expected doublets prior to multiplexing and identification of doublets based on SNP-information with SoupLadle should robustly identify and even reduce the amount of doublets in snRNA-data comparison to standard workflows.

Importantly, we believe that the major advantage of SoupLadle compared to the more established labeling approaches (Hashtagging or Cellplex) is that our protocol does not add additional steps during nuclei isolation. In contrast, standard labeling approaches such as Hashtagging or Cellplex recommend three washing steps after cell/nuclei labeling. As each washing step leads to a loss of ~20-30% of nuclei (<https://www.biorxiv.org/content/10.1101/2020.10.23.351809v1.full>), this leads to an inevitable loss of >50% of samples, which is detrimental for processing of precious biopsy samples. In contrast, our protocol enables immediate sample pooling after initial nuclei isolation, and thus not only improves nuclei loss but also significantly simplifies sample processing, as all samples can be further processed and FACS-sorted within a single tube. In summary, we believe that faster nuclei isolation and processing, better overall cell assignment, and less cell type bias make SoupLadle the best multiplexing approach for precious biopsy samples.

We have adapted the manuscript as follows to include (1) the two new multiplexing datasets, and (2) the discussion of the advantages and disadvantages of SoupLadle multiplexing. Changes to the manuscript are marked in gray.

*“Next, we decided to benchmark SoupLadle in a larger snRNA-seq dataset (n=8, 11501 nuclei) isolated from frozen human heart tissue (**Figure 2f, Extended Data Fig. 2i-k**). Of note, due to excessive nuclei loss with hashtag or CMO labeling as a consequence of the required processing and washing steps, we did not perform cell labeling. Importantly, as both SoupLadle and Vireo outperformed cell labeling approaches in our previous benchmark (**Figure 1d**), the latter cannot be considered a viable benchmark for SNP multiplexing tools. In line with our previous in-silico analysis (**Figure 1k**), SoupLadle showed a robust performance with higher sample numbers, assigning 99% of cells to patients. In contrast, Vireo Vartrix bulkRNA cell assignment dropped with higher sample numbers to 81% assigned cells, while Vireo CellSNP bulkRNA only assigned 45% of cells (**Figure 2g**). Assessing the overlap in patient assigned cells across multiplexing methods showed a 100% overlap between*

SoupLadle and Vireo Vartrix bulkRNA and a 96% overlap of SoupLadle and Vireo CellSNP bulkRNA, corroborating a robust cell assignment by SoupLadle and Vireo (**Extended Data Fig. 2l-m**). In line with our hypothesis that consideration of all SNPs in snRNA data is critical for the successful assignment of less abundant cell-types, SoupLadle showed less cell-type bias, with better demultiplexing of rare cell-types (e.g. Neuronal cluster) in comparison to both Vireo Vartrix bulkRNA and Vireo CellSNP bulkRNA (**Figure 2h**).

To further validate our approach in (1) tissue characterized by a high cell heterogeneity and (2) smaller tissue samples, we next tested multiplexing of frozen human kidney biopsies (n=4, 11490 nuclei) (**Figure 2i, Extended Data Fig. 2n-p**). Supporting our previous results, SoupLadle showed the highest cell assignment with 96% of cells assigned, followed by a drop in Vireo Vartrix bulkRNA demultiplexing performance with only 21% of cells assigned, while Vireo CellSNP bulkRNA failed to adequately demultiplex samples (<1% assigned). Again, overlap of assigned cells was high (~93%) in SoupLadle and Vireo Vartrix, while Vireo CellSNP did not adequately assign cells in comparison to other methods (**Extended Data Fig. 2r-s**). Analysis of kidney tissue strongly highlighted a Vireo Vartrix celltype bias, with inefficient demultiplexing of rare cell-types with distinct transcript expression such as podocytes, fibroblasts, and intercalated cells, in comparison to tubular cells (PT, TAL & DCT), which represent the most common cell-type in the kidney (**Figure 2j**). In contrast, SoupLadle showed no apparent cell-type bias, highlighting the advantage of our novel multiplexing approach (**Figure 2j**)."

"Comparing our novel approach to standard scRNA-seq costs, SoupLadle provides a ~4-fold cost reduction (even higher when performing bulkRNA-seq without rRNA-depletion) when multiplexing eight samples (**Supplemental Table 1**). More importantly, one of the key advantages of SoupLadle in comparison to standard cell-labeling approaches (Hashtaging, Cellplex) is that it does not require additional experimental steps during cell isolation. Instead, samples can be pooled immediately after cell or nuclei isolation, without having to label and wash samples prior to pooling. This reduces processing steps, which is critical for isolation of nuclei, as the latter are particularly sensitive to disruption or nuclei loss during washing steps (~20-30% of nuclei are lost with each washing step)¹⁰. In contrast to standard Cell-Labeling approaches, SoupLadle requires bulkRNA-seq or WES. ... While theoretically bulkRNA-seq can be performed in parallel to cell or nuclei isolation, we recommend an a priori evaluation of SNP heterogeneity (**SoupLadle Workflow, Figure 1h**). This additional step enables optimal selection of samples for pooling based on bulkRNA SNPs and avoids critical loss of samples in multiplexed scRNA/snRNA-data due to insufficient SNP heterogeneity. As such, SoupLadle is best suited for frozen tissue, where bulkRNA-seq can be performed prior to multiplexing from a small piece (e.g. 1 mg) of frozen tissue."

...

To assess doublet identification of multiplexing methods in more detail we next assessed the shared doublet assignment for each method including manually and computationally assigned doublets (scDbfFinder⁹) using computationally identified doublets by scDbfFinder as a reference (**Extended Data Fig. 1g**). In line with our prior observation, Vireo CellSNP bulkRNA and WES showed the lowest overlap with scDbfFinder (**Figure 1f**) as both methods underassign doublets (**Extended Data Fig. 1g**). In contrast, Soupcell showed the

highest overlap with both manually and computationally inferred doublets by scDbiFinder, highlighting its robust doublet assignment.

...

Online Methods

Nuclei isolation from snap-frozen tissue

“... The above steps for Hashtagging and CMO-Labeling were not performed for the larger heart dataset, where eight samples were pooled, and the kidney biopsy dataset, as the additional processing steps led to excessive nuclei loss with insufficient remaining nuclei remaining for adequate chip loading (loading > 10000 cells on the chip).”

Doublet estimation

For doublet estimation from single-cell RNA-seq data, we applied scDbiFinder⁹, which simulates artificial doublets by computationally merging transcriptomes from pairs of randomly selected cells. We computed the doublet score for the top 4000 genes and determined cells with a score > 3 as doublet. Subsequently the overlapping assignments with all applied methods and manual cluster annotation were evaluated using scDbiFinder-identified doublets as a reference.

2. What are the cost advantages when whole genome sequencing (\$500?) is required to generate enough SNP's for demultiplexing successfully? When pooling only 5 samples, cost advantages might not justify problems arising from pooling strategies.

The authors thank the reviewer for this critical question and now provide an excel file, with which users can calculate cost advantages (**New Supplemental Table 1**). At default the authors have estimated costs for multiplexing n=8 samples (this number was performed successfully within the revision). Here the cost-advantages would be a 4-fold reduction of costs per sample from originally 1930€ to 484€ per sample (**New Supplemental Table 1**). More importantly, the authors want to emphasize that our strategy will not only reduce costs, but for the first time enable processing of large batches in parallel. In contrast to standard labeling approaches, where samples are isolated separately, labeled with HTO-Antibodies or CMO, washed three times, and only then pooled, our approach needs no additional steps during cell/nuclei isolation, allowing early pooling of samples during isolation. It is important to mention that both established cell-labeling approaches such as Hashtagging or Cellplex recommend three washes after cell labeling. However as each nuclei washing step is associated with a ~20-30% nuclei loss this inevitably leads to a >50% (0,7³-0,8³) sample loss with cell labeling approaches (<https://www.biorxiv.org/content/10.1101/2020.10.23.351809v1.full>), which is detrimental for the processing of small and rare human biopsy samples. In contrast, SoupLadle enables researchers to isolate large sample numbers in parallel, e.g. allowing isolation of n=8 samples simultaneously, which we deem critical for generation of large cohort snRNA-seq data (n > 100). We have adapted the manuscript as follows to reflect this. Changes to the manuscript are marked in gray.

“Comparing our novel approach to standard scRNA-seq costs, SoupLadle provides a ~4-fold cost reduction (even higher when performing bulkRNA-seq without rRNA-depletion) when multiplexing eight samples (**Supplemental Table 1**). More importantly, one of the key

*advantages of SoupLadle in comparison to standard cell-labeling approaches (Hashtagging, Cellplex) is that it does not require additional experimental steps during cell isolation. Instead, samples can be pooled immediately after cell or nuclei isolation, without having to label and wash samples prior to pooling. This reduces processing steps, which is critical for isolation of nuclei, as the latter are particularly sensitive to disruption or nuclei loss during washing steps (~20-30% of nuclei are lost with each washing step)¹⁰. In contrast to standard Cell-Labeling approaches, SoupLadle requires bulkRNA-seq or WES. ... While theoretically bulkRNA-seq can be performed in parallel to cell or nuclei isolation, we recommend an a priori evaluation of SNP heterogeneity (**SoupLadle Workflow, Figure 1h**). This additional step enables optimal selection of samples for pooling based on bulkRNA SNPs and avoids critical loss of samples in multiplexed scRNA/snRNA-data due to insufficient SNP heterogeneity. As such, SoupLadle is best suited for frozen tissue, where bulkRNA-seq can be performed prior to multiplexing from a small piece (e.g. 1 mg) of frozen tissue.”*

3. The authors need to provide also absolute values for cells assigned to the specific sample origin not only percentages. How many cells from each sample has been submitted? How many cells have been recovered? This will show whether equal numbers can be obtained from each sample or whether there are difficulties/variabilities with pooling even with PBMC's.

The authors agree with the reviewer, that it is critical to not only consider percentages, but also absolute cell numbers. For PBMC we sorted 100000 cells per patient and pooled samples immediately after sorting. For heart tissue we sorted whole samples for dataset 1 (due to extensive nuclei loss due to cell labeling with both Cellplex and Hashtags). For the heart dataset 2 we sorted approximately equal nuclei numbers per patient (50000 nuclei). Samples were pooled immediately after sorting. For kidney biopsies we pooled total processed single cell suspensions of biopsies prior to sorting to minimize tissue loss. In addition, we have provided all information on cell sorting numbers, assignment and cell recovery for all methods and datasets as total cell numbers including average number of cells per patient, and the standard deviation of cells recovered per patient (**New Supplemental Table 2**). For the PBMC dataset SoupPorcell/SoupLadle assigned 17858 of 17892 cells (Pat 1: 3507, Pat 2: 3241, Pat 3: 4178, Pat 4: 2572, Pat 5: 1987, Doublets 2373). Similarly, SoupLadle assigned 1532 of 1563 cells (Pat 6: 468, Pat 7: 207, Pat 8: 419, Pat 9: 179, Pat 10: 176, Doublets: 83 cells) for the first snRNA-seq heart dataset. Similar numbers were achieved for SoupLadle in the novel heart and kidney datasets (**New Supplemental Table 2**), where differences in performance between SoupLadle and Vireo Vartrix bulkRNA and Vireo CellSNP bulkRNA are even more apparent.

Importantly, while there is an inherent variability in cells recovered per patient despite equal sorting numbers for each patient (standard deviation of 24-44% across datasets for SoupLadle), the authors believe that this is rather a result of wetlab processing inaccuracies (e.g. varying sorting efficiency, differing times of sorted samples in tubes due to sequential sorting of samples, pipetting), than a biased computational multiplexing algorithm, as virtually all cells are assigned to patients (96-99% for SoupPorcell/SoupLadle) (**New Supplemental Table 2**). In addition, SoupLadle shows a lower standard deviation in cells recovered per patient than Vireo CellSNP, Vireo Vartrix, and Hashtag multiplexing, as well as similar numbers to Cellplex multiplexing, underscoring it's efficiency in unbiased cell demultiplexing

(**New Supplemental Table 2**). Lastly, as mentioned above, SoupLadle shows the least cell-type bias across methods, demultiplexing both abundant and rare cell-types to patients in an unbiased manner (**Figure 2e, h and j**).

We have adapted the manuscript as follows. Changes to the manuscript are marked in gray.

Online Methods

PBMC isolation

“...For each sample we sorted an equal number of cells (100,000 cells per sample). After sorting, PBMC were pooled and immediately loaded onto a Chromium Next GEM Chip G for snRNA-seq (10x, 3' v3.1) with a cell recovery of ~18000 cells after Cell Ranger alignment. ...”

Nuclei isolation from snap-frozen tissue

“...For the first heart dataset complete samples were sorted due to a low amount of recovered nuclei due to the additional required processing steps for Cellplex and Hashtag labeling (exact numbers provided in Supplemental Table 2), while for the second heart dataset an approximately equal number of nuclei per sample was sorted (~50000 nuclei per patient) and subsequently pooled. For kidney biopsies, where tissue was scarce, total processed biopsies were pooled prior to sorting to reduce sample loss. ...”

Single-cell RNA-seq demultiplexing

*“All absolute numbers for cell assignment are provided in **Supplemental Table 2**.”*

4. Which QC parameters (cutoffs) have been applied to identify the transcriptome for a cell? (How many genes per cell?). How many cells have been targeted (per run/ per sample?)

Cells were aligned with default CellRanger/Seurat settings (CellRanger-7.1.0 and reference genome GRCh38-2020-A) for cell identification, but otherwise no QC parameters (e.g. mitochondrial fraction cutoffs, feature cutoffs) were applied prior to demultiplexing. We reasoned that any form of quality control, such as the exclusion of cells with high mitochondrial read fraction, high/low features, or high/low counts should be minimized prior to demultiplexing, as it may confound subsequent demultiplexing results and benchmarks. In our workflow, we suggest first to perform demultiplexing, and subsequently QC. Indeed, we believe that demultiplexing will aid in setting correct QC cutoffs as doublets identified via SoupLadle/Vireo/Hashtag/CMO demultiplexing can be used as a reference for setting the upper cutoffs for cell features and counts. The exact numbers of cells recovered after CellRanger alignment are now provided in a **New Supplemental Table 2**. We have adjusted the manuscript as follows to reflect this (changes are marked in gray).

Online Methods

“For scRNA-/snRNA-seq alignment CellRanger was run with standard settings (CellRanger-7.1.0 and reference genome GRCh38-2020-A) for cell identification, but otherwise no QC cutoffs were set prior to demultiplexing, as the latter may confound demultiplexing analysis. Samples were subsequently analyzed with Seurat. First, samples were normalized and scaled (NormalizeData, ScaleData). For clustering we

used 4000 highly variable genes to compute principal component analysis, and subsequent UMAP dimensionality reduction. Clustering was performed at the lowest resolution (FindClusters, res. 0.1) to identify major clusters.”

...

*“All absolute numbers for cell assignment are provided in **Supplemental Table 2.**”*

5. The authors showed that exome sequencing is insufficient to identify enough SNPs which can be used for the demultiplexing procedure after executing 3'RNAseq experiments. But many researchers use also 5' RNAseq kits. The authors may be able to calculate by using publicly available 5' data whether cheaper exome sequencing could be an alternative to WGS.

As suggested by the reviewer, the authors now assess SNP distribution in 5' scRNA-seq samples of a publicly available 5' scRNA-seq dataset (PMID: 35959412). As expected, we recovered a comparable number of SNPs to 3' scRNA-seq and 3' snRNA-seq datasets (**New Extended Data Fig. 3a, Figure 1c, Extended Data Fig. 2g**). Similar to our prior observation in 3' scRNA-seq datasets, the majority of recovered SNPs were located within introns, as well as the 5' and 3' UTRs (**New Extended Data Fig. 3b**). Of note, in comparison to our previous analyses we did not assess overlap of scRNA SNPs with bulkRNA/WES SNPs, as we did not find a suitable 5' scRNA-seq dataset with patient-specific bulkRNA and WES data for overlap analysis. However, in previous analyses we demonstrate that SNPs located within introns, the 3' UTR and 5' UTR are better recovered with bulkRNA-seq than WES (**Figure 1e and 2c**). As such, our novel analysis suggests that 5' scRNA-seq multiplexing could be feasible with SoupLadle, as a comparable amount of SNPs are quantified in 5' and 3' scRNA-seq data, and that bulkRNA-seq would be better suited for 5' scRNA-seq multiplexing than WES, as bulkRNA-seq maps introns, 3' and 5' UTRs better (**Figure 1e, 2c**). We have adapted the manuscript as follows to reflect this (changes are marked in gray).

*“Lastly, as we analyzed SoupLadle multiplexing performance in 3' scRNA-seq datasets only, we aimed to assess whether SoupLadle could be suitable for 5' scRNA-seq multiplexing. Here we analyzed SNP distribution in a published 5' scRNA-seq dataset (**Extended Data Fig. 3a**)⁹. In line with our 3' scRNA-seq analyses, we recovered a comparable amount of SNPs and found that the majority of recovered SNPs were located within introns, as well as 5' and 3' UTRs (**Extended Data Fig. 3a-b, Figure 1c, Extended Data Fig. 2g**). These results suggest that bulkRNA-seq would be best suitable for 5' scRNA-seq multiplexing, as the latter better captures SNPs located within introns and UTRs than WES.”*

6. Ambient RNA is huge problem in scRNA transcriptics. There are bioinformatics tools to partially remove those in case of unlikely appearance in wrong cell types. When pooling various patient samples ambient RNA will show up in the wrong patient and might mislead downstream analysis and results and might be more difficult to remove with current tools. Have the authors thought about using their genetic markers to identify the amount of ambient RNA and how to remove those from the datasets?

The authors agree with the reviewer that ambient RNA is a major problem in scRNA and particularly snRNA datasets. We did not evaluate ambient RNA removal here due to the lack of a “positive control” or definitive knowledge on the amount of ambient RNA contamination across experiments, that would allow us to accurately benchmark different algorithms. However, SoupLadle is compatible with ambient RNA detection algorithms

provided within SoupORcell. As such it is possible to simultaneously deconvolute cells and detect/remove ambient RNA with SoupLadle leveraging SoupORcell. More importantly, pooling patients with distinct SNP profiles should improve ambient RNA detection, as ambient RNA can be more accurately identified based on patient-unique SNPs detected in ambient transcripts (as contaminating transcripts within a droplet will often be from other patients and therefore express distinct SNPs). In summary, while we do not provide a proprietary solution for ambient RNA detection, SoupLadle can be used with standard detection algorithms for ambient RNA, and specifically SoupORcell, which infers ambient RNA based on patient-specific SNPs. We have adapted the manuscript as follows to reflect this (changes are marked in gray).

“Of note, while we do not provide a novel tool for ambient RNA correction, ambient RNA can be imputed with standard packages (CellBender, SoupX) or SoupORcell, which leverages the natural genetic variation of multiplexed patients to estimate ambient RNA.”

Minor:

7. In online methods: please explain why 25,000 reads /cell have been used to sequence the scRNAseq libraries but only 2500 reads per cell for the CellPlex and Hashtag libraries! Is the poor performance of these label-free methods explainable with low sequence coverage?

Different sequencing depth are recommended for (3') scRNA gene expression libraries and Hashtag/Cellplex libraries, as the complexity of Hashtag and Cellplex libraries (in our case only 5 distinct Hashtag sequences or 5 distinct CMOs oligo sequences) is much lower than that of 3' RNA libraries, where the goal is to capture the expression of the whole genome. The targeted sequencing depth of 2500 reads/cell reflects the recommendation by Stoeckius et al. for Hashtag-sequencing libraries of ~ 2k reads/cell (<https://cite-seq.com/protocols/>). For the PBMC dataset we obtained a median of 28,676 reads per cell for the gene expression library and a median of 3,380 CMO UMIs per cell. To clarify this the authors have adapted the methods section as follows (changes are marked in gray).

“For CellPlex and Hashtag libraries we targeted a sequencing depth of 2500 reads per cell based on recommendations for sequencing depth of Hashtag/Cite-seq libraries (<https://cite-seq.com/>).”

8. In Main: Authors talk about “genetic multiplexing methods” and probably mixed it with “labelling methods” in this context because “genetic” implies the availability of SNP information.

The authors thank the reviewer for pointing out this error and have corrected the manuscript as follows (changes are marked in gray).

“At baseline however (without reference WES or bulkRNA-seq SNP data), SNP-calling methods only discriminate (from here on referred to as deconvolute), but don't re-assign cells to patients due to the lack of information on reference patient-defining SNPs.”

9. In Methods: Probably a typo? usually 250 million sequence reads rather than 25 million sequence read depth used for sequencing each library (in case 10000 cells are targeted).

As the reviewer indicates this is a misunderstanding. We performed bulkRNA-sequencing with a read-depth of 25 Mio reads per sample, which is a standard-sequencing depth for bulkRNA samples. For scRNA-seq samples, we sequenced samples at a read depth of 25000 reads / cell, which translates to 250 Mio reads per 10000 cells targeted. To clarify this, we have adapted the online methods section as follows (changes are marked in gray).

Single cell/nuclear, CellPlex and Hashtag library preparation

“After quality control on an Agilent TapeStation, scRNA/snRNA-seq samples were sequenced on an Illumina NovaSeq system targeting a sequencing depth of 25000 reads/cell.”

Bulk-RNA library preparation, sequencing, alignment and SNP-calling

“...After quality control (Agilent TapeStation), bulkRNA-seq samples were sequenced on an Illumina NovaSeq system targeting 25 million reads/bulkRNA library.”

10. References: the following manuscript should be cited: “Multiplexed droplet single-cell RNA-sequencing using natural genetic variation”, Hyun Min Kang et al., Nature Biotechnology volume 36, pages89–94 (2018).

The authors thank the reviewer for pointing out this missed reference in the manuscript and have now included this reference as follows (changes are marked in gray).

“Alternatively, cells can be demultiplexed by calling patient-specific SNPs from scRNA-seq data (Vireo, Souporcell or Demuxlet)⁴⁻⁸.”

Reviewer #2 (Remarks on code availability):

I am not a bioinformatician and not equipped to read code.

REVIEWERS' COMMENTS

Reviewer #1 (Remarks to the Author):

The authors have sufficiently addressed most of my feedback and have better demonstrated the performance of SoupLadle versus other demultiplexing methods like Vireo CellSNP. New analysis shown in Figure 1k is very helpful to show that the performance advantage is not driven by more SNPs alone.

Remaining minor concerns include:

- UMAPs in stacked format are still essentially conceptual cartoons (perhaps this is the goal and should be explicitly stated) rather than conveying experimental results.
- Fig 1h: There seem to be errors in the matching between the cartoon cells and patients?
- Fig 1f: I am confused about the correlations shown in this figure suggesting max distance between matching samples from the two methods. The color scale labels might be reversed?

Reviewer #2 (Remarks to the Author):

The authors addressed all my concerns satisfactorily. I was particularly impressed by their novel multiplexing experiments, which demonstrated their strategy's success in complex tissues, such as human kidney biopsies.

Additionally, they showed that multiplexing more samples is feasible, reducing costs further. This advancement is significant for the cell biology field, making crucial single-cell level experiments more accessible to more labs. Experiments which were previously prohibitively expensive. The authors provided now an excel file which allows calculating cost advantages regarding various pooling strategies. This is very much appreciated.

All my questions regarding method details, 5'scRNAseq chemistry, ambient RNA challenges, and missing references have also been very satisfyingly addressed.

This is a very impressive and thoroughly generated rebuttal indeed!

The authors have answered all remaining comments and would like to thank the reviewers for their careful assessment that significantly improved our manuscript.

Reviewer #1 (Remarks to the Author):

The authors have sufficiently addressed most of my feedback and have better demonstrated the performance of SoupLadle versus other demultiplexing methods like Vireo CellSNP. New analysis shown in Figure 1k is very helpful to show that the performance advantage is not driven by more SNPs alone.

Remaining minor concerns include:

- UMAPs in stacked format are still essentially conceptual cartoons (perhaps this is the goal and should be explicitly stated) rather than conveying experimental results.

The authors have adapted the manuscript to explicitly state that these are conceptual cartoons as suggested by the reviewer. We have adapted the Figure Legends as follows (changes are highlighted in gray):

Figure Legends

Figure 1: ... b, Conceptual cartoon showing UMAP representations of PBMC stratified by clustering or patient assignment of multiplexing methods.

Figure 2: a, Conceptual cartoon showing UMAP representations of cardiac nuclei ... f, Conceptual cartoon showing UMAP representations of cardiac nuclei (dataset 2, n=8) ... i, Conceptual cartoon showing UMAP representations of kidney nuclei (dataset 3, n=4)...

- Fig 1h: There seem to be errors in the matching between the cartoon cells and patients?

The authors thank the reviewer for pointing out this error and have now adapted the cartoon in **Figure 1h** to correct this.

- Fig 1f: I am confused about the correlations shown in this figure suggesting max distance between matching samples from the two methods. The color scale labels might be reversed?

The authors thank the reviewer for pointing out this error. Indeed, the color scale labels were switched. We have corrected this error in the new **Figure 1j**.

Reviewer #2 (Remarks to the Author):

The authors addressed all my concerns satisfactorily. I was particularly impressed by their novel multiplexing experiments, which demonstrated their strategy's success in complex tissues, such as human kidney biopsies.

Additionally, they showed that multiplexing more samples is feasible, reducing costs further. This advancement is significant for the cell biology field, making crucial single-cell level experiments more accessible to more labs. Experiments which were previously prohibitively

expensive. The authors provided now an excel file which allows calculating cost advantages regarding various pooling strategies. This is very much appreciated. All my questions regarding method details, 5'scRNAseq chemistry, ambient RNA challenges, and missing references have also been very satisfyingly addressed. This is a very impressive and thoroughly generated rebuttal indeed!